# Broadcast Product: Redefining Shape-aligned Element-wise Multiplication and Beyond

**Yusuke Matsui**                                                    *matsui@hal.t.u-tokyo.ac.jp*
*The University of Tokyo*

**Tatsuya Yokota**                                                    *t.yokota@nitech.ac.jp*
*Nagoya Institute of Technology*
*RIKEN Center for Advanced Intelligence Project*

**Reviewed on OpenReview:** *https://openreview.net/forum?id=zv00tOPpPO*

## Abstract

Broadcast operations are widely used in scientific computing libraries, yet their mathematical formulation is often implicit and inconsistently represented in machine learning literature. This problem frequently leads to invalid equations when element-wise products are written despite mismatched tensor shapes. In this paper, we formalize such operations by introducing the *broadcast product* ⊡, which explicitly extends the Hadamard product through shape-aligned element duplication. We provide a rigorous definition of the broadcast product, analyze its algebraic properties, and show how it can be expressed using standard linear algebra. Building on this framework, we formulate least-squares problems and sketch a proof-of-concept *broadcast decomposition*. As a preliminary illustration, we show that the formalism enables a new family of decompositions with distinct structural properties from conventional tensor decompositions. This work establishes a mathematical foundation for broadcast-aware tensor operations, connecting practical implementations with rigorous tensor analysis.

## 1 Introduction

Broadcast operations are a fundamental abstraction in scientific computing libraries for performing mathematical operations on tensors of different shapes. They automatically replicate elements as needed to align tensor shapes before applying the specified operation, enabling concise and expressive tensor computations. For example, in libraries such as `numpy` (Harris et al., 2020), adding a vector `x` to a matrix `A` implicitly replicates `x` to match the shape of `A`. Moreover, languages such as `MATLAB` and `Julia` (Bezanson et al., 2026) support broadcasting at the language level.

Although we cannot directly write broadcast operations into equations, many modern machine learning papers have attempted to do so by mimicking `numpy`'s descriptions. For example, consider selecting a region of a three-channel image $\boldsymbol{\mathcal{X}} \in \mathbb{R}^{H \times W \times 3}$ using a binary mask $\boldsymbol{B} \in \{0,1\}^{H \times W}$ to obtain $\boldsymbol{\mathcal{Y}} \in \mathbb{R}^{H \times W \times 3}$, where $H$ and $W$ denote the height and width, respectively. One often describes this operation by mimicking a `numpy`-style expression, `Y = X * B`, and reformulating it using the Hadamard product $\odot$ as follows:

$$\boldsymbol{\mathcal{Y}} = \boldsymbol{\mathcal{X}} \odot \boldsymbol{B}. \tag{1}$$

However, this expression is incorrect because the Hadamard product is only defined for tensors of the same shape. Therefore, this equation implicitly assumes the mask is broadcast along the channel dimension. Such incorrect notation not only creates discrepancies between the source code and mathematical expressions but also leads to invalid mathematical manipulations and flawed discussions.

For example, for any two tensors with the same shape $\boldsymbol{\mathcal{M}}$ and $\boldsymbol{\mathcal{N}}$, the $L_0$ norm of their Hadamard product must be smaller than that of the original tensor: $\|\boldsymbol{\mathcal{M}} \odot \boldsymbol{\mathcal{N}}\|_0 \le \|\boldsymbol{\mathcal{N}}\|_0$. This is because the result of the

Hadamard product can only increase the number of zero elements, never decrease it. However, substituting Equation (1) into this inequality gives $\|\boldsymbol{\mathcal{Y}}\|_0 = \|\boldsymbol{\mathcal{X}} \odot \boldsymbol{B}\|_0 \leq \|\boldsymbol{B}\|_0$, which is clearly incorrect[1]. In this way, improper use of broadcast operations can lead to mathematically flawed reasoning. Many people tend to write this masking formula, and it has even appeared in award-nominated papers at top conferences (e.g., Figure 5 in Li et al. (2023b) and Equation 2 in Lin et al. (2021))[2].

To address this issue, we redefine the *broadcast product* operator, denoted by $\boxdot$. This operator extends the Hadamard product $\odot$ by duplicating elements to align tensor shapes before multiplication. When tensor shapes are given explicitly (i.e., all axis sizes are specified), it is mathematically equivalent to broadcasting in `numpy`; the difference arises only when tensor orders differ, due to the distinction between the F-convention and C-convention (formally introduced in Definition 1). The masking operation above can be expressed as:

$$\boldsymbol{\mathcal{Y}} = \boldsymbol{\mathcal{X}} \boxdot \boldsymbol{B} = \boldsymbol{\mathcal{X}} \odot \boldsymbol{\mathcal{B}}^{\square}. \tag{2}$$

Here, $\boldsymbol{\mathcal{B}}^{\square} \in \{0,1\}^{H \times W \times 3}$ denotes the result of duplicating and concatenating the mask $\boldsymbol{B}$ along the channel dimension (as detailed later in Definition 3). Now, the inequality discussed above can be correctly written as: $\|\boldsymbol{\mathcal{Y}}\|_0 = \|\boldsymbol{\mathcal{X}} \boxdot \boldsymbol{B}\|_0 = \|\boldsymbol{\mathcal{X}} \odot \boldsymbol{\mathcal{B}}^{\square}\|_0 \leq \|\boldsymbol{\mathcal{B}}^{\square}\|_0 = 3\|\boldsymbol{B}\|_0$. The proposed operator $\boxdot$ can accurately describe complex problems, potentially offering new interpretations of the problem and possibilities for optimization from alternative perspectives.

Furthermore, we analyze the mathematical properties of the broadcast product and position it within the context of tensor decomposition. We first represent the broadcast product in graphical notation widely used in tensor decomposition (Penrose, 1971; Taylor, 2024; Yokota, 2024). Next, we approximate an arbitrary tensor as the broadcast product of two tensors and minimize the reconstruction error, proposing a solution to the least squares problem based on the broadcast product. This least-squares formulation allows us to decompose an arbitrary tensor into multiple tensors via broadcast products, which we term *broadcast decomposition.*

Our contributions are summarized as follows:

- **Notation/formalization contribution:** We propose a notation for explicitly representing broadcast-aware element-wise operations. This fills the gap between informal `numpy`-style descriptions and rigorous mathematical notation.

- **Analytical contribution:** Under this notation, we provide a systematic definition of the broadcast product, summarize its basic algebraic properties, and clarify its relationship to existing operations such as the Hadamard product, Khatri–Rao product, and standard tensor decompositions.

- **Proof-of-concept decomposition:** Building on this framework, we sketch a broadcast decomposition (BD) as a preliminary result rather than a central contribution. We do not claim empirical superiority; rather, we show that BD has distinct structural properties from conventional tensor decompositions.

Following Kolda & Bader (2009), scalars, vectors, matrices, and tensors are denoted by $x \in \mathbb{R}$, $\boldsymbol{x} \in \mathbb{R}^I$, $\boldsymbol{X} \in \mathbb{R}^{I \times J}$, and $\boldsymbol{\mathcal{X}} \in \mathbb{R}^{I \times J \times K}$, respectively. The element-wise (Hadamard) product is denoted by $\boldsymbol{\mathcal{X}} \odot \boldsymbol{\mathcal{Y}}$, and the element-wise division is denoted by $\boldsymbol{\mathcal{X}} \oslash \boldsymbol{\mathcal{Y}}$. An element of a tensor $\boldsymbol{\mathcal{X}} \in \mathbb{R}^{I \times J \times K}$ is written as $x_{ijk} \in \mathbb{R}$. For a third-order tensor $\boldsymbol{\mathcal{X}} \in \mathbb{R}^{I \times J \times K}$, the frontal slice ($k$-th channel) is denoted by $\boldsymbol{X}_k \in \mathbb{R}^{I \times J}$. For an $N$-th order tensor $\boldsymbol{\mathcal{X}} \in \mathbb{R}^{I_1 \times I_2 \times \cdots \times I_N}$ and a matrix $\boldsymbol{A} \in \mathbb{R}^{J \times I_n}$, we denote the $n$-th mode tensor-matrix product by $\boldsymbol{\mathcal{X}} \times_n \boldsymbol{A} \in \mathbb{R}^{I_1 \times \cdots \times I_{n-1} \times J \times I_{n+1} \times \cdots \times I_N}$ (we retain the term "mode" in tensor-decomposition terminology, such as the $n$-th mode product $\times_n$ and mode-$n$ unfolding, following Kolda & Bader (2009)). We assume the underlying field to be $\mathbb{R}$, although the definitions extend straightforwardly to $\mathbb{C}$.

---

[1]For example, consider a $1 \times 1$ three-channel image $\boldsymbol{\mathcal{X}} \in \mathbb{R}^{1 \times 1 \times 3}$ and a mask $\boldsymbol{B} \in \{0,1\}^{1 \times 1}$, where all elements of $\boldsymbol{\mathcal{X}}$ and $\boldsymbol{B}$ are non-zero ($\|\boldsymbol{B}\|_0 = 1$). In this case, if we apply the incorrect Equation (1), all three elements of $\boldsymbol{\mathcal{Y}}$ also become non-zero, meaning that we have $\|\boldsymbol{\mathcal{Y}}\|_0 = 3$. Thus, the inequality $\|\boldsymbol{\mathcal{Y}}\|_0 \leq \|\boldsymbol{B}\|_0$, which should always hold, is no longer satisfied.

[2]We emphasize that this point concerns notation only and does not affect the proposed algorithms.

Figure 1: Example of the broadcast product of a third-order tensor $\boldsymbol{\mathcal{X}} \in \mathbb{R}^{3 \times 4 \times 2}$ and a matrix $\boldsymbol{Y} \in \mathbb{R}^{3 \times 4}$. $\boldsymbol{\mathcal{X}}^{\square}, \boldsymbol{\mathcal{Y}}^{\square} \in \mathbb{R}^{3 \times 4 \times 2}$ represent their broadcasted forms. $\boldsymbol{X}_{\square}, \boldsymbol{Y}_{\square} \in \mathbb{R}^{3 \times 4}$ illustrate their Frobenius-norm marginalization.

## 1.1 Examples

Before going into the detailed definition, let us introduce the broadcast product intuitively. If two input tensors have identical shapes, their broadcast product equals their Hadamard product, e.g., with $\boldsymbol{x}, \boldsymbol{y} \in \mathbb{R}^2$:

$$\boldsymbol{x} \boxdot \boldsymbol{y} = \begin{bmatrix} 1 \\ 2 \end{bmatrix} \boxdot \begin{bmatrix} 3 \\ 4 \end{bmatrix} = \begin{bmatrix} 1 \\ 2 \end{bmatrix} \odot \begin{bmatrix} 3 \\ 4 \end{bmatrix} = \begin{bmatrix} 3 \\ 8 \end{bmatrix}. \tag{3}$$

Next, we show an example of matrices with different shapes, $\boldsymbol{X} \in \mathbb{R}^{3 \times 2}$ and $\boldsymbol{y} \in \mathbb{R}^{1 \times 2}$, as follows:

$$\boldsymbol{X} \boxdot \boldsymbol{y} = \begin{bmatrix} 1 & 2 \\ 3 & 4 \\ 5 & 6 \end{bmatrix} \boxdot \begin{bmatrix} 7 & 8 \end{bmatrix} = \begin{bmatrix} 1 & 2 \\ 3 & 4 \\ 5 & 6 \end{bmatrix} \odot \begin{bmatrix} 7 & 8 \\ 7 & 8 \\ 7 & 8 \end{bmatrix} = \begin{bmatrix} 7 & 16 \\ 21 & 32 \\ 35 & 48 \end{bmatrix}. \tag{4}$$

Here, to match the shapes on both sides of $\boxdot$, $\boldsymbol{y}$ is duplicated along the first axis, producing a matrix with the same shape as $\boldsymbol{X}$ (shown in red). The Hadamard product with $\boldsymbol{X}$ is then computed. Note that although the result coincides with the Khatri–Rao product (Kolda & Bader, 2009) in this case (the Khatri–Rao product $\boldsymbol{B} \odot_{\mathrm{KR}} \boldsymbol{A}$ concatenates column-wise Kronecker products of two matrices with the same number of columns), the broadcast product and the Khatri–Rao product are generally different. For instance, the Khatri–Rao product is defined for $\boldsymbol{X} \in \mathbb{R}^{3 \times 2}$ and $\boldsymbol{Y} \in \mathbb{R}^{2 \times 2}$, whereas the broadcast product is not (see Section 6.1 for details).

Next, let us look at an example of the product of a third-order tensor $\boldsymbol{\mathcal{X}} \in \mathbb{R}^{3 \times 4 \times 2}$ and a matrix $\boldsymbol{Y} \in \mathbb{R}^{3 \times 4}$:

$$\boldsymbol{X}_1 = \begin{bmatrix} 1 & 4 & 7 & 10 \\ 2 & 5 & 8 & 11 \\ 3 & 6 & 9 & 12 \end{bmatrix}, \quad \boldsymbol{X}_2 = \begin{bmatrix} 13 & 16 & 19 & 22 \\ 14 & 17 & 20 & 23 \\ 15 & 18 & 21 & 24 \end{bmatrix}, \quad \boldsymbol{Y} = \begin{bmatrix} -1 & 2 & 3 & 4 \\ -5 & 6 & 7 & 8 \\ -9 & 10 & 11 & 12 \end{bmatrix}. \tag{5}$$

$\boldsymbol{\mathcal{X}} \boxdot \boldsymbol{Y}$ is equivalent to duplicating $\boldsymbol{Y}$ along the third axis to match shapes and computing the Hadamard product. This example closely resembles the masking operation discussed in Equation (2), as illustrated in the upper part of Figure 1; the notation of $\boldsymbol{\mathcal{X}}^{\square}$ and $\boldsymbol{\mathcal{Y}}^{\square}$ is formally defined in Section 2.

## 2 Definition

Let us define the broadcast product in detail. We first introduce two conventions for identifying tensors of different orders.

> **Definition 1** (F-convention and C-convention). *When tensors of different orders are compared, a shape-normalization convention is needed to equalize their orders by inserting singleton axes.*
>
> ***F-convention*** *(Fortran-style): trailing singleton axes are appended. For example, $\mathbb{R}^{3\times4}$ is identified with $\mathbb{R}^{3\times4\times1}$ (and recursively with $\mathbb{R}^{3\times4\times1\times1}$, etc.). In particular, writing $\boldsymbol{x}\in\mathbb{R}^3$ denotes a column vector, i.e., $\boldsymbol{x}\in\mathbb{R}^{3\times1}$. Mathematical notation commonly used in machine learning and computer vision, as well as* `Julia`, *follow this convention.*
>
> ***C-convention*** *(C-style): leading singleton axes are prepended. For example, $\mathbb{R}^{3\times4}$ is identified with $\mathbb{R}^{1\times3\times4}$.* `numpy` *and* `pytorch` *follow this convention.*
>
> *Throughout this paper we adopt the F-convention. See Appendix G for a detailed comparison and concrete examples of the discrepancy between the two formats.*

Following Definition 1, we define the *broadcast condition*, which determines whether the broadcast operation can be applied to two tensors.

> **Definition 2** (Broadcast Condition). *Under the F-convention (Definition 1), two tensors $\boldsymbol{\mathcal{X}}$ and $\boldsymbol{\mathcal{Y}}$ are first brought to the same order $N$ by appending trailing singleton axes as needed (e.g., $\boldsymbol{\mathcal{Y}}\in\mathbb{R}^{I_1\times\cdots\times I_M}$ with $M<N$ becomes $\boldsymbol{\mathcal{Y}}\in\mathbb{R}^{I_1\times\cdots\times I_M\times1\times\cdots\times1}$). Writing the resulting shapes as $\boldsymbol{\mathcal{X}}\in\mathbb{R}^{I_1\times\cdots\times I_N}$ and $\boldsymbol{\mathcal{Y}}\in\mathbb{R}^{J_1\times\cdots\times J_N}$, the pair $(\boldsymbol{\mathcal{X}},\boldsymbol{\mathcal{Y}})$ satisfies the* broadcast condition *if for every $n\in\{1,2,\ldots,N\}$, at least one of the following holds: (1) $I_n=J_n$, (2) $I_n=1$, or (3) $J_n=1$.*

For example, the following meet the broadcast condition.

- Same shape: $\boldsymbol{\mathcal{X}}\in\mathbb{R}^{3\times2},\boldsymbol{\mathcal{Y}}\in\mathbb{R}^{3\times2}$.

- The length of an axis ($J_2$) is one: $\boldsymbol{\mathcal{X}}\in\mathbb{R}^{3\times2},\boldsymbol{\mathcal{Y}}\in\mathbb{R}^{3\times1}$.

- The lengths of some axes are one ($I_1=J_2=1$): $\boldsymbol{\mathcal{X}}\in\mathbb{R}^{1\times2\times5},\boldsymbol{\mathcal{Y}}\in\mathbb{R}^{3\times1\times5}$.

- Multiple ones: $\boldsymbol{\mathcal{X}}\in\mathbb{R}^{1\times1\times5},\boldsymbol{\mathcal{Y}}\in\mathbb{R}^{3\times1\times5}$.

- Different orders (F-convention): $\boldsymbol{\mathcal{X}}\in\mathbb{R}^{5\times4\times3}$, $\boldsymbol{\mathcal{Y}}\in\mathbb{R}^{5\times4}$ (identified with $\mathbb{R}^{5\times4\times1}$).

The following do not satisfy the broadcast condition.

- The length of an axis is different and not one ($I_2=2,J_2=3$): $\boldsymbol{\mathcal{X}}\in\mathbb{R}^{3\times2},\boldsymbol{\mathcal{Y}}\in\mathbb{R}^{3\times3}$.

- Different orders with first-axis mismatch: $\boldsymbol{\mathcal{X}}\in\mathbb{R}^{3\times2}$, $\boldsymbol{\mathcal{Y}}\in\mathbb{R}^{4\times2\times5}$ (under F-convention, $\boldsymbol{\mathcal{X}}$ is identified with $\mathbb{R}^{3\times2\times1}$; since $I_1=3\neq4=J_1$ and neither equals 1, the condition fails at axis 1).

Next, we define the *broadcast product* as follows.

> **Definition 3** (Broadcast Product). *Let $\boldsymbol{\mathcal{X}}$ and $\boldsymbol{\mathcal{Y}}$ be two tensors satisfying the broadcast condition (Definition 2), with their shapes brought to the same order $N$ under the F-convention (Definition 1). Writing $\boldsymbol{\mathcal{X}}\in\mathbb{R}^{I_1\times\cdots\times I_N}$ and $\boldsymbol{\mathcal{Y}}\in\mathbb{R}^{J_1\times\cdots\times J_N}$, the broadcast product $\boldsymbol{\mathcal{Z}}=\boldsymbol{\mathcal{X}}\boxdot\boldsymbol{\mathcal{Y}}$ is defined as follows, where $\boldsymbol{\mathcal{Z}},\boldsymbol{\mathcal{X}}^\square,\boldsymbol{\mathcal{Y}}^\square\in\mathbb{R}^{\max(I_1,J_1)\times\cdots\times\max(I_N,J_N)}$.*
>
> $$\boldsymbol{\mathcal{Z}}=\boldsymbol{\mathcal{X}}\boxdot\boldsymbol{\mathcal{Y}}:=\mathrm{bc}(\boldsymbol{\mathcal{X}},\mathrm{size}(\boldsymbol{\mathcal{Y}}))\odot\mathrm{bc}(\boldsymbol{\mathcal{Y}},\mathrm{size}(\boldsymbol{\mathcal{X}}))=\boldsymbol{\mathcal{X}}^\square\odot\boldsymbol{\mathcal{Y}}^\square.$$

Here, "size" returns the input tensor's shape as a tuple, e.g., $\mathrm{size}(\boldsymbol{\mathcal{Y}})=(J_1,J_2,\ldots,J_N)$, and "bc" is a function to duplicate a tensor. For $\boldsymbol{\mathcal{X}}$ and $\boldsymbol{\mathcal{Y}}$ satisfying the broadcast condition, bc inputs $\boldsymbol{\mathcal{X}}$ itself and the shape of $\boldsymbol{\mathcal{Y}}$, and outputs $\boldsymbol{\mathcal{X}}^\square$, i.e., $\boldsymbol{\mathcal{X}}^\square=\mathrm{bc}(\boldsymbol{\mathcal{X}},\mathrm{size}(\boldsymbol{\mathcal{Y}}))$. Here, $\boldsymbol{\mathcal{X}}^\square$ refers to the broadcasted version of $\boldsymbol{\mathcal{X}}$, which means the following. For all $n$, if $I_n=1$, replicate all elements of $\boldsymbol{\mathcal{X}}$ along the $n$-th axis $J_n$ times. This operation is explicitly defined using index notation as follows. With $k_n\in\{1,2,\ldots,\max(I_n,J_n)\}$ for all $n$, we write the $(k_1,k_2,\ldots,k_N)$-th element of $\boldsymbol{\mathcal{X}}^\square$ as $x^\square_{k_1k_2\ldots k_N}=x_{i_1'i_2'\ldots i_N'}$, where

$$i_n'=\begin{cases}1 & (I_n=1),\\ k_n & (I_n\neq1).\end{cases} \tag{6}$$

The same applies to $\boldsymbol{\mathcal{Y}}^{\square}$. In the end, we obtain

$$z_{k_1,k_2,\dots,k_N} = x^{\square}_{k_1 k_2 \dots k_N} y^{\square}_{k_1 k_2 \dots k_N} = x_{i'_1 i'_2 \dots i'_N} y_{j'_1 j'_2 \dots j'_N}, \quad \text{where} \quad j'_n = \begin{cases} 1 & (J_n = 1), \\ k_n & (J_n \neq 1). \end{cases} \tag{7}$$

In other words, placing $\square$ on the superscript means duplicating the elements appropriately so that the shapes of $\boldsymbol{\mathcal{X}}^{\square}$ and $\boldsymbol{\mathcal{Y}}^{\square}$ match, i.e., size($\boldsymbol{\mathcal{X}}^{\square}$) = size($\boldsymbol{\mathcal{Y}}^{\square}$). Note that $\boldsymbol{\mathcal{X}}^{\square}$ and $\boldsymbol{\mathcal{Y}}^{\square}$ are considered as a shorthand notation, and should only be used when the interpretation is unique and obvious from the context.

The result of the broadcast product is uniquely determined. In the example of Equation (4), we can write:

$$\boldsymbol{X}^{\square} = \boldsymbol{X} = \begin{bmatrix} 1 & 2 \\ 3 & 4 \\ 5 & 6 \end{bmatrix}, \quad \boldsymbol{Y}^{\square} = \begin{bmatrix} \boldsymbol{y} \\ \boldsymbol{y} \\ \boldsymbol{y} \end{bmatrix} = \begin{bmatrix} 7 & 8 \\ 7 & 8 \\ 7 & 8 \end{bmatrix}. \tag{8}$$

In the example of Equation (5) and Figure 1, we obtain

$$\boldsymbol{\mathcal{X}}^{\square} = \boldsymbol{\mathcal{X}}, \quad \boldsymbol{Y}_1^{\square} = \boldsymbol{Y}_2^{\square} = \boldsymbol{Y} = \begin{bmatrix} -1 & 2 & 3 & 4 \\ -5 & 6 & 7 & 8 \\ -9 & 10 & 11 & 12 \end{bmatrix}, \tag{9}$$

where stacking $\boldsymbol{Y}_1^{\square}$ and $\boldsymbol{Y}_2^{\square}$ along the third axis leads to $\boldsymbol{\mathcal{Y}}^{\square}$. The following is an example of duplication for both $\boldsymbol{\mathcal{X}}$ and $\boldsymbol{\mathcal{Y}}$. Let us define $\boldsymbol{\mathcal{X}} \in \mathbb{R}^{1 \times 2 \times 3}$ and $\boldsymbol{\mathcal{Y}} \in \mathbb{R}^{4 \times 2 \times 1}$:

$$\boldsymbol{X}_1 = [1, 2], \quad \boldsymbol{X}_2 = [3, 4], \quad \boldsymbol{X}_3 = [5, 6], \quad \boldsymbol{Y}_1 = \begin{bmatrix} 7 & 8 \\ 9 & 10 \\ 11 & 12 \\ 13 & 14 \end{bmatrix}. \tag{10}$$

In this case, $\boldsymbol{\mathcal{X}}^{\square}, \boldsymbol{\mathcal{Y}}^{\square} \in \mathbb{R}^{4 \times 2 \times 3}$ are written as

$$\boldsymbol{X}_1^{\square} = \begin{bmatrix} 1 & 2 \\ 1 & 2 \\ 1 & 2 \\ 1 & 2 \end{bmatrix}, \quad \boldsymbol{X}_2^{\square} = \begin{bmatrix} 3 & 4 \\ 3 & 4 \\ 3 & 4 \\ 3 & 4 \end{bmatrix}, \quad \boldsymbol{X}_3^{\square} = \begin{bmatrix} 5 & 6 \\ 5 & 6 \\ 5 & 6 \\ 5 & 6 \end{bmatrix}, \quad \boldsymbol{Y}_1^{\square} = \boldsymbol{Y}_2^{\square} = \boldsymbol{Y}_3^{\square} = \begin{bmatrix} 7 & 8 \\ 9 & 10 \\ 11 & 12 \\ 13 & 14 \end{bmatrix}. \tag{11}$$

An equivalent and always valid alternative is to write $\boldsymbol{\mathcal{X}}^{\square} \odot \boldsymbol{\mathcal{Y}}^{\square}$ explicitly, which may be clearer when the broadcast structure itself is the focus; $\square$ is preferable when the same pattern recurs across equations, since it attaches the broadcasting semantics to the operator rather than the operands.

## 3 Properties

We present various mathematical properties of the broadcast product. These properties help convert mathematical expressions using the broadcast product into the standard forms used in linear algebra.

### 3.1 Basic properties

We first show the basic properties of the broadcast product $\square$.

**Compatibility**: When $\boldsymbol{\mathcal{X}}$ and $\boldsymbol{\mathcal{Y}}$ have the same shapes, $\square$ is equivalent to $\odot$.

$$\boldsymbol{\mathcal{X}} \square \boldsymbol{\mathcal{Y}} = \boldsymbol{\mathcal{X}} \odot \boldsymbol{\mathcal{Y}}. \tag{12}$$

**Commutativity**: For $\boldsymbol{\mathcal{X}}$, $\boldsymbol{\mathcal{Y}}$, and $\boldsymbol{0}$ satisfying the broadcast conditions, the following holds, where $k \in \mathbb{R}$.

$$\boldsymbol{\mathcal{X}} \square \boldsymbol{\mathcal{Y}} = \boldsymbol{\mathcal{Y}} \square \boldsymbol{\mathcal{X}}. \tag{13}$$

$$(k\boldsymbol{\mathcal{X}}) \boxdot \boldsymbol{\mathcal{Y}} = \boldsymbol{\mathcal{X}} \boxdot (k\boldsymbol{\mathcal{Y}}) = k(\boldsymbol{\mathcal{X}} \boxdot \boldsymbol{\mathcal{Y}}). \tag{14}$$

$$\boldsymbol{\mathcal{X}} \boxdot \boldsymbol{0} = \boldsymbol{0} \boxdot \boldsymbol{\mathcal{X}} = \boldsymbol{0}. \tag{15}$$

**Associativity**: When $\boldsymbol{\mathcal{X}}$, $\boldsymbol{\mathcal{Y}}$, and $\boldsymbol{\mathcal{Z}}$ pairwise satisfy the broadcast conditions, we obtain

$$(\boldsymbol{\mathcal{X}} \boxdot \boldsymbol{\mathcal{Y}}) \boxdot \boldsymbol{\mathcal{Z}} = \boldsymbol{\mathcal{X}} \boxdot (\boldsymbol{\mathcal{Y}} \boxdot \boldsymbol{\mathcal{Z}}) = \boldsymbol{\mathcal{X}} \boxdot \boldsymbol{\mathcal{Y}} \boxdot \boldsymbol{\mathcal{Z}}. \tag{16}$$

Note that, even if $(\boldsymbol{\mathcal{X}}, \boldsymbol{\mathcal{Y}})$ and $(\boldsymbol{\mathcal{X}}, \boldsymbol{\mathcal{Z}})$ satisfy the broadcast conditions, $(\boldsymbol{\mathcal{Y}}, \boldsymbol{\mathcal{Z}})$ do not necessarily satisfy the broadcast conditions, e.g., $\boldsymbol{\mathcal{X}} \in \mathbb{R}^{3 \times 1}$, $\boldsymbol{\mathcal{Y}} \in \mathbb{R}^{3 \times 2}$, and $\boldsymbol{\mathcal{Z}} \in \mathbb{R}^{3 \times 5}$.

**Distributivity**: When $\boldsymbol{\mathcal{Y}}$ and $\boldsymbol{\mathcal{Z}}$ have identical shapes, and $\boldsymbol{\mathcal{X}}$ and $\boldsymbol{\mathcal{Y}}$ satisfy the broadcast condition,

$$\boldsymbol{\mathcal{X}} \boxdot (\boldsymbol{\mathcal{Y}} + \boldsymbol{\mathcal{Z}}) = \boldsymbol{\mathcal{X}} \boxdot \boldsymbol{\mathcal{Y}} + \boldsymbol{\mathcal{X}} \boxdot \boldsymbol{\mathcal{Z}}. \tag{17}$$

Even if $(\boldsymbol{\mathcal{X}}, \boldsymbol{\mathcal{Y}})$ and $(\boldsymbol{\mathcal{X}}, \boldsymbol{\mathcal{Z}})$ satisfy the broadcast conditions, differing shapes of $\boldsymbol{\mathcal{Y}}$ and $\boldsymbol{\mathcal{Z}}$ make the left-hand side uncomputable, though the right-hand side may be; e.g., $\boldsymbol{\mathcal{X}} \in \mathbb{R}^{2 \times 3}$, $\boldsymbol{\mathcal{Y}} \in \mathbb{R}^{2 \times 1}$, and $\boldsymbol{\mathcal{Z}} \in \mathbb{R}^{1 \times 3}$.

**Generalization of the Hadamard Product**: Whenever the Hadamard product is well-defined (i.e., the operands have identical shapes), it can be replaced with the broadcast product. That is, the equation $\boldsymbol{\mathcal{A}} \odot \boldsymbol{\mathcal{B}}$ can always be rewritten as $\boldsymbol{\mathcal{A}} \boxdot \boldsymbol{\mathcal{B}}$. This is because the use of the Hadamard product implies that $\boldsymbol{\mathcal{A}}$ and $\boldsymbol{\mathcal{B}}$ have the same shape, allowing it to be rewritten as the broadcast product via Equation (12). Note that, however, the Hadamard product is often misused as discussed in Equation (1).

## 3.2 Frobenius-norm Marginalization and the Frobenius norm

We introduce Frobenius-norm marginalization for the broadcast product, enabling the efficient computation of the Frobenius norm and the derivation of the Cauchy–Schwarz inequality. We first define Frobenius-norm marginalization formally. Here, we write : to denote the full index range of an axis, following `MATLAB`/`Julia` convention (Golub & Van Loan, 2013). For example, $\boldsymbol{X}_{:,j}$ is the $j$-th column of $\boldsymbol{X}$.

---

**Definition 4** (Frobenius-norm Marginalization)**.** *Let $\boldsymbol{\mathcal{X}}$ and $\boldsymbol{\mathcal{Y}}$ be two tensors satisfying the broadcast condition (Definition 2), with their shapes brought to the same order $N$ under the F-convention (Definition 1). Writing $\boldsymbol{\mathcal{X}} \in \mathbb{R}^{I_1 \times I_2 \times \cdots \times I_N}$ and $\boldsymbol{\mathcal{Y}} \in \mathbb{R}^{J_1 \times J_2 \times \cdots \times J_N}$, the marginalized tensors are written as*

$$\boldsymbol{\mathcal{X}}_\square, \boldsymbol{\mathcal{Y}}_\square \in \mathbb{R}^{\min(I_1,J_1) \times \min(I_2,J_2) \times \cdots \times \min(I_N,J_N)}.$$

*These are defined using index notation. With $k_n \in \{1, 2, \ldots, \min(I_n, J_n)\}$ for all $n$,*

$$x_{\square k_1 k_2 \ldots k_N} = \|\boldsymbol{\mathcal{X}}_{\bar{i}_1 \bar{i}_2 \ldots \bar{i}_N}\|_F, \quad \bar{i}_n = \begin{cases} : & (J_n = 1), \\ k_n & (J_n \neq 1). \end{cases}$$

---

That is, the marginalized tensor $\boldsymbol{\mathcal{X}}_\square$ is obtained by taking the Frobenius norm along each axis of $\boldsymbol{\mathcal{X}}$ if the axis's length is longer than that of $\boldsymbol{\mathcal{Y}}$. Here, $\boldsymbol{\mathcal{Y}}_\square$ is defined similarly. This Frobenius-norm marginalization (hereafter simply *marginalization*) can also be regarded as a process that shrinks the shape of a tensor while preserving the Frobenius norm. Analogously to marginalizing a probability distribution (which preserves the $L_1$ norm), our operation reduces a tensor by one or more axes while preserving its Frobenius norm. As is clear from the construction, we have $\|\boldsymbol{\mathcal{X}}\|_F = \|\boldsymbol{\mathcal{X}}_\square\|_F \leq \|\boldsymbol{\mathcal{X}}^\square\|_F$ in general. Also, all elements of the marginalized tensor are non-negative: $\boldsymbol{\mathcal{X}}_\square \geq \boldsymbol{0}$. In the example of Equation (4), we obtain

$$\boldsymbol{x}_\square = \begin{bmatrix} \sqrt{35} & \sqrt{56} \end{bmatrix}, \quad \boldsymbol{y}_\square = \begin{bmatrix} 7 & 8 \end{bmatrix}. \tag{18}$$

Figure 1 shows the example of Equation (5). As this example of $\boldsymbol{Y}$ shows, negative elements must become positive even if the shape does not change. For Equation (10), we obtain

$$\boldsymbol{x}_\square = \begin{bmatrix} \sqrt{35} & \sqrt{56} \end{bmatrix}, \quad \boldsymbol{y}_\square = \begin{bmatrix} \sqrt{420} & \sqrt{504} \end{bmatrix}. \tag{19}$$

Using the marginalized tensors, we can write the Frobenius norm of the broadcast product as follows:

$$\|\boldsymbol{\mathcal{X}} \boxdot \boldsymbol{\mathcal{Y}}\|_F = \|\boldsymbol{\mathcal{X}}^\square \odot \boldsymbol{\mathcal{Y}}^\square\|_F = \|\boldsymbol{\mathcal{X}}_\square \odot \boldsymbol{\mathcal{Y}}_\square\|_F. \tag{20}$$

With this, one can compute the Frobenius norm efficiently. Computing the norm requires "enlarging" the tensors via $\boldsymbol{\mathcal{X}}^{\square}$, but one can compute it using smaller tensors by first "shrinking" them via $\boldsymbol{\mathcal{X}}_{\square}$ (see Figure 1).

From this, we can derive that the Cauchy–Schwarz inequality also holds for the broadcast product:

**Theorem 3.1** (Cauchy–Schwarz Inequality for the Broadcast Product)**.** *Let $\boldsymbol{\mathcal{X}}$ and $\boldsymbol{\mathcal{Y}}$ be tensors satisfying the broadcast conditions. Then:*

$$\|\boldsymbol{\mathcal{X}} \boxdot \boldsymbol{\mathcal{Y}}\|_F \leq \|\boldsymbol{\mathcal{X}}\|_F \cdot \|\boldsymbol{\mathcal{Y}}\|_F.$$

*Proof.* This follows directly from Equation (20) and the standard Cauchy–Schwarz inequality. See Appendix A for the complete proof. □

### 3.3 Properties of lower-order tensors

We describe the properties for lower-order tensors. By employing these transformations, one can express the description of the broadcast product using standard linear algebra notation.

**Scalar**: For any tensor $\boldsymbol{\mathcal{X}}$ and a scalar $y \in \mathbb{R}$, we obtain

$$\boldsymbol{\mathcal{X}} \boxdot y = y\boldsymbol{\mathcal{X}}. \tag{21}$$

**Vector and vector**: For vectors $\boldsymbol{x}, \boldsymbol{y} \in \mathbb{R}^I$ of equal length, the following holds:

$$\boldsymbol{x} \boxdot \boldsymbol{y} = \boldsymbol{x} \odot \boldsymbol{y} = \mathrm{diag}(\boldsymbol{x})\boldsymbol{y} \in \mathbb{R}^I. \tag{22}$$

$$\boldsymbol{x} \boxdot \boldsymbol{y}^\top = \boldsymbol{x}\boldsymbol{y}^\top \in \mathbb{R}^{I \times I}. \tag{23}$$

For vectors $\boldsymbol{x} \in \mathbb{R}^I$ and $\boldsymbol{y} \in \mathbb{R}^J$ of different lengths, $\boldsymbol{x} \boxdot \boldsymbol{y}$ cannot be defined, but $\boldsymbol{x} \boxdot \boldsymbol{y}^\top$ can be defined:

$$\boldsymbol{x} \boxdot \boldsymbol{y}^\top = \boldsymbol{x}\boldsymbol{y}^\top \in \mathbb{R}^{I \times J}. \tag{24}$$

**Matrix and vector**: For a matrix $\boldsymbol{X} \in \mathbb{R}^{I \times J}$, vectors $\boldsymbol{y} \in \mathbb{R}^I$ and $\boldsymbol{z} \in \mathbb{R}^J$, the following holds:

$$\boldsymbol{X} \boxdot \boldsymbol{y} = \boldsymbol{X} \odot [\boldsymbol{y}, \boldsymbol{y}, \ldots, \boldsymbol{y}] = \boldsymbol{X} \odot (\boldsymbol{y}\boldsymbol{1}_J^\top) = \mathrm{diag}(\boldsymbol{y})\boldsymbol{X}. \tag{25}$$

$$\boldsymbol{X} \boxdot \boldsymbol{z}^\top = \boldsymbol{X} \odot \begin{bmatrix} \boldsymbol{z}^\top \\ \vdots \\ \boldsymbol{z}^\top \end{bmatrix} = \boldsymbol{X} \odot (\boldsymbol{1}_I \boldsymbol{z}^\top) = \boldsymbol{X}\mathrm{diag}(\boldsymbol{z}). \tag{26}$$

Additionally, the following holds for the Frobenius norm:

$$\|\boldsymbol{X} \boxdot \boldsymbol{y}\|_F^2 = \|\mathrm{diag}(\boldsymbol{y})\boldsymbol{X}\|_F^2 = \mathrm{tr}(\boldsymbol{X}^\top \mathrm{diag}^2(\boldsymbol{y})\boldsymbol{X}). \tag{27}$$

**Third-order tensor and matrix**: For a third-order tensor $\boldsymbol{\mathcal{X}} \in \mathbb{R}^{I \times J \times K}$ and a matrix $\boldsymbol{Y} \in \mathbb{R}^{I \times J}$, the following holds. Here, $\mathrm{fold}_n$ denotes the inverse of mode-$n$ unfolding; see Appendix B for the definition and a visual illustration. Figure 1 is a visual example.

$$\boldsymbol{\mathcal{X}} \boxdot \boldsymbol{Y} = \boldsymbol{\mathcal{X}} \odot \mathrm{fold}_1(\boldsymbol{1}_K^\top \otimes \boldsymbol{Y}) = \mathrm{fold}_3(\boldsymbol{X}_{(3)}\mathrm{diag}(\mathrm{vec}(\boldsymbol{Y}))). \tag{28}$$

This operation is typical in computer vision, e.g., $\boldsymbol{\mathcal{X}}$ represents an image and $\boldsymbol{Y}$ serves as a grayscale mask.

### 3.4 Graphical notations

Here, we show how to draw the broadcast product in graphical notations of tensors (Penrose, 1971; Taylor, 2024; Yokota, 2024). Here, we draw tensors as nodes with multiple edges, represented by black circles and lines. Additionally, we draw super-diagonal tensors using slashed circles, $\oslash$, instead of black circles for arbitrary tensors. An $N$-th order tensor $\boldsymbol{\mathcal{I}} \in \mathbb{R}^{I \times \cdots \times I}$ is called *super-diagonal* if its $(k_1, k_2, \ldots, k_N)$-th

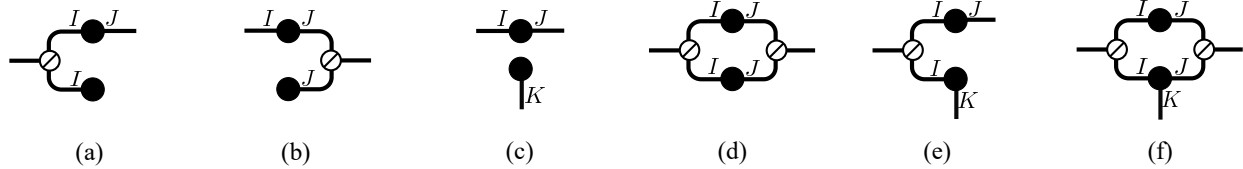

Figure 2: Graphical notations of various patterns of broadcast product: Diagrams (a)-(f) represent broadcast products between $(I, J)$-matrix with (a) $(I, 1)$-vector, (b) $(1, J)$-vector, (c) $(1, 1, K)$-vector, (d) $(I, J)$-matrix, (e) $(I, 1, K)$-matrix, and (f) $(I, J, K)$-tensor, respectively. Especially, case (c) is equivalent to outer product, and case (d) equivalent to Hadamard product.

element equals 1 when $k_1 = k_2 = \cdots = k_N$ and 0 otherwise; this generalises the identity matrix to higher orders. For example, the third-order super-diagonal tensor $\boldsymbol{\mathcal{I}} \in \mathbb{R}^{3 \times 3 \times 3}$ is given by

$$\boldsymbol{I}_1 = \begin{bmatrix} 1 & 0 & 0 \\ 0 & 0 & 0 \\ 0 & 0 & 0 \end{bmatrix}, \quad \boldsymbol{I}_2 = \begin{bmatrix} 0 & 0 & 0 \\ 0 & 1 & 0 \\ 0 & 0 & 0 \end{bmatrix}, \quad \boldsymbol{I}_3 = \begin{bmatrix} 0 & 0 & 0 \\ 0 & 0 & 0 \\ 0 & 0 & 1 \end{bmatrix}. \tag{29}$$

Figure 2 shows the graphical notation of various patterns of the broadcast product. In this way, the broadcast product can be represented as a product (network) of tensors via super-diagonal tensors.

Figure 2a represents a case of Equation (25). Since a tensor product of a super-diagonal tensor with a vector is a diagonal matrix

$$\boldsymbol{\mathcal{I}} \times_3 \boldsymbol{y}^\top = \mathrm{diag}(\boldsymbol{y}), \tag{30}$$

the property Equation (25) can be immediately verified. The property Equation (26) can also be verified in a similar way. Figure 2f represents a case of Equation (28). For the proof, see Appendix B. Although Penrose graphical notation (Penrose, 1971) has not yet been widely adopted in mainstream machine learning, it is the standard language in tensor-network communities spanning computational physics (Ran et al., 2020) and machine learning (Stoudenmire & Schwab, 2016; Cichocki et al., 2016; 2017; Li et al., 2023a; Núñez Fernández et al., 2025). We adopt it here because it makes the axis structure and contractions visually explicit, enabling an intuitive understanding of the broadcast product and broadcast decomposition in Section 5.2.

### 3.5 Broadcast sum, difference, and division

The broadcast sum ($\boxplus$), difference ($\boxminus$), and division ($\boxslash$) are similarly defined for $\boldsymbol{\mathcal{X}}$ and $\boldsymbol{\mathcal{Y}}$ that meet the broadcast condition:

$$\boldsymbol{\mathcal{X}} \boxplus \boldsymbol{\mathcal{Y}} := \boldsymbol{\mathcal{X}}^\square + \boldsymbol{\mathcal{Y}}^\square, \qquad \boldsymbol{\mathcal{X}} \boxminus \boldsymbol{\mathcal{Y}} := \boldsymbol{\mathcal{X}}^\square - \boldsymbol{\mathcal{Y}}^\square, \qquad \boldsymbol{\mathcal{X}} \boxslash \boldsymbol{\mathcal{Y}} := \boldsymbol{\mathcal{X}}^\square \oslash \boldsymbol{\mathcal{Y}}^\square, \tag{31}$$

where $\boldsymbol{\mathcal{Y}}$ must not have zero elements for $\boxslash$. See Appendix C for the basic properties of these operators. See Appendix F for LaTeX commands and PowerPoint input methods for these operators.

## 4 Real-world Examples

This section demonstrates that the broadcast product naturally formalizes a wide range of operations commonly described using informal or ambiguous notation in machine learning. Through the broadcast operations, various expressions can be represented in a unified manner, offering new perspectives on them.

### 4.1 Blending

Similar to the masking operation in Equation (2), multi-channel blending can be expressed using the broadcast product. Consider two color images $\boldsymbol{\mathcal{X}}, \boldsymbol{\mathcal{Y}} \in \mathbb{R}^{H \times W \times 3}$ with height $H$ and width $W$ and a mask

$A \in [0,1]^{H \times W}$ representing an alpha channel. The blending operation is written as

$$\boldsymbol{\mathcal{X}} \boxdot \boldsymbol{A} + \boldsymbol{\mathcal{Y}} \boxdot (\mathbf{1} - \boldsymbol{A}), \tag{32}$$

where $\mathbf{1}$ denotes an all-ones matrix of the same shape as $\boldsymbol{A}$. This formulation naturally extends to batched inputs $\boldsymbol{\mathcal{X}}, \boldsymbol{\mathcal{Y}} \in \mathbb{R}^{N \times H \times W \times C}$, where $N$ denotes the batch size, by defining the mask as $\boldsymbol{\mathcal{A}} \in [0,1]^{1 \times H \times W}$ (or equivalently $[0,1]^{1 \times H \times W \times 1}$), and applying Equation (32) directly.

## 4.2 Batch Normalization

Batch Normalization (Ioffe & Szegedy, 2015), Layer Normalization (Ba et al., 2016), and Instance Normalization (Ulyanov et al., 2016) can all be described in a unified manner using broadcast operations. We first demonstrate that the following commonly used notation is not mathematically rigorous:

$$\mathrm{BN}(x) = \left( \frac{x - \mu}{\sigma} \right) \gamma + \beta. \tag{33}$$

Here, $x \in \mathbb{R}^{N \times C \times H \times W}$ denotes the input tensor, where $N$, $C$, $H$, and $W$ are the batch size, number of channels, height, and width, respectively, following the channel-second convention of Wu & He (2018). Here, $\mu, \sigma \in \mathbb{R}^C$ denote the batch-wise mean and standard deviation, and $\gamma, \beta \in \mathbb{R}^C$ are learnable parameters.

This expression is mathematically incorrect, as tensor operations with different shapes rely on implicit broadcasting, e.g., we cannot divide a tensor by a vector ($\sigma$).

Let us now rewrite Batch Normalization explicitly using index notation. Let the input and output tensors be $\boldsymbol{\mathcal{X}}, \boldsymbol{\mathcal{Y}} \in \mathbb{R}^{N \times C \times H \times W}$. Batch Normalization is defined as follows. For each channel $c \in \{1, 2, \ldots, C\}$, let us define the mean $\mu_c \in \mathbb{R}$ and standard deviation $\sigma_c \in \mathbb{R}$ by:

$$\mu_c = \frac{1}{NHW} \sum_{n=1}^{N} \sum_{h=1}^{H} \sum_{w=1}^{W} X_{n,c,h,w}, \quad \sigma_c = \sqrt{\frac{1}{NHW} \sum_{n=1}^{N} \sum_{h=1}^{H} \sum_{w=1}^{W} (X_{n,c,h,w} - \mu_c)^2 + \varepsilon}. \tag{34}$$

Here, $\varepsilon$ is a small constant introduced to prevent division by zero. Using the channel-wise learned parameters $\gamma_1, \ldots, \gamma_C \in \mathbb{R}$ and $\beta_1, \ldots, \beta_C \in \mathbb{R}$, Batch Normalization can be written element-wise as:

$$Y_{n,c,h,w} = \left( \frac{X_{n,c,h,w} - \mu_c}{\sigma_c} \right) \gamma_c + \beta_c. \tag{35}$$

This representation is rigorous. However, because it focuses on individual elements, the notation becomes cluttered with indices. The resulting tensor $\boldsymbol{\mathcal{Y}}$ cannot be easily substituted into other equations.

By contrast, using the broadcast operations, Equation (35) can be concisely expressed at the tensor level as:

$$\boldsymbol{\mathcal{Y}} = ((\boldsymbol{\mathcal{X}} \boxminus \boldsymbol{\mu}) \boxslash \boldsymbol{\sigma}) \boxdot \boldsymbol{\gamma} \boxplus \boldsymbol{\beta}. \tag{36}$$

Here, $\boldsymbol{\mu} \in \mathbb{R}^{1 \times C \times 1 \times 1}$ is the tensor containing $\mu_1, \ldots, \mu_C$, and $\boldsymbol{\sigma}, \boldsymbol{\gamma}, \boldsymbol{\beta} \in \mathbb{R}^{1 \times C \times 1 \times 1}$ are defined similarly.

Under the F-convention (Definition 1), trailing singleton axes can be omitted; for example, $\boldsymbol{\mu} \in \mathbb{R}^{1 \times C \times 1 \times 1}$ can be written as $\boldsymbol{\mu} \in \mathbb{R}^{1 \times C}$. For clarity and consistency, we explicitly specify all shapes in the upcoming normalization examples.

## 4.3 Layer Normalization (for CNN)

Next, let us consider Layer Normalization (Ba et al., 2016). Here, we follow the notation convention used in Wu & He (2018). Let the input tensor be $\boldsymbol{\mathcal{X}} \in \mathbb{R}^{N \times C \times H \times W}$. In Layer Normalization for CNNs, the mean over all elements of each instance $n \in \{1, \ldots, N\}$ is defined as $\mu_n = \frac{1}{CHW} \sum_{c=1}^{C} \sum_{h=1}^{H} \sum_{w=1}^{W} X_{n,c,h,w}$. The standard deviation is defined as $\sigma_n = \sqrt{\frac{1}{CHW} \sum_{c=1}^{C} \sum_{h=1}^{H} \sum_{w=1}^{W} (X_{n,c,h,w} - \mu_n)^2 + \varepsilon}$. Stacking these values gives $\boldsymbol{\mu}, \boldsymbol{\sigma} \in \mathbb{R}^{N \times 1 \times 1 \times 1}$. As in Batch Normalization, the per-channel learnable parameters are $\boldsymbol{\gamma}, \boldsymbol{\beta} \in \mathbb{R}^{1 \times C \times 1 \times 1}$. The output tensor is computed using the same expression as in Equation (36).

### 4.4 Layer Normalization (for Transformer)

Layer Normalization is also widely used in Transformer models (Phuong & Hutter, 2022), but its usage differs slightly from the CNN case. Let the input tensor be $\mathcal{X} \in \mathbb{R}^{N \times L \times D}$, where $N$ is the batch size, $L$ is the number of tokens, and $D$ is the embedding dimension. Unlike the CNN case, where normalization is performed over all elements of an instance, Transformer-style Layer Normalization computes statistics only over the embedding dimension. For each $n \in \{1, \ldots, N\}$ and $l \in \{1, \ldots, L\}$, the mean is defined as $\mu_{n,l} = \frac{1}{D} \sum_{d=1}^{D} X_{n,l,d}$. The standard deviation is defined as $\sigma_{n,l} = \sqrt{\frac{1}{D} \sum_{d=1}^{D} (X_{n,l,d} - \mu_{n,l})^2 + \varepsilon}$. Stacking these values gives $\boldsymbol{\mu}, \boldsymbol{\sigma} \in \mathbb{R}^{N \times L \times 1}$. In this setting, the learnable parameters are defined per embedding dimension as $\boldsymbol{\gamma}, \boldsymbol{\beta} \in \mathbb{R}^{1 \times 1 \times D}$. The output tensor is again computed using the same expression in Equation (36).

### 4.5 Instance Normalization

Finally, let us consider Instance Normalization (Ulyanov et al., 2016). Let the input tensor be $\mathcal{X} \in \mathbb{R}^{N \times C \times H \times W}$. The mean over the spatial dimensions is defined as $\mu_{n,c} = \frac{1}{HW} \sum_{h=1}^{H} \sum_{w=1}^{W} X_{n,c,h,w}$. The standard deviation is defined as $\sigma_{n,c} = \sqrt{\frac{1}{HW} \sum_{h=1}^{H} \sum_{w=1}^{W} (X_{n,c,h,w} - \mu_{n,c})^2 + \varepsilon}$. Stacking these values gives $\boldsymbol{\mu}, \boldsymbol{\sigma} \in \mathbb{R}^{N \times C \times 1 \times 1}$. With the learnable parameters $\boldsymbol{\gamma}, \boldsymbol{\beta} \in \mathbb{R}^{1 \times C \times 1 \times 1}$, the output of Instance Normalization is computed using the same expression Equation (36) as for Batch Normalization and Layer Normalization.

In this way, Batch, Layer, and Instance Normalization can all be written in a unified and mathematically rigorous form by appropriately choosing the shapes of the variables. In contrast, the commonly used abbreviated notation in Equation (33) cannot precisely express these relationships and introduces ambiguity. For example, the fact that $\sigma$ is a vector rather than a scalar is not apparent from that notation alone.

### 4.6 AdaIN

Adaptive Instance Normalization (AdaIN) (Huang & Belongie, 2017), widely used in style transfer, can also be expressed in the form of Equation (36).

Let the input tensor be $\mathcal{X} \in \mathbb{R}^{N \times C \times H \times W}$. The transferred style information is represented by channel-wise affine parameters, which are given as $\boldsymbol{\gamma}, \boldsymbol{\beta} \in \mathbb{R}^{1 \times C \times 1 \times 1}$. As in standard Instance Normalization, the mean and standard deviation of the input tensor are defined per instance and per channel, yielding $\boldsymbol{\mu}, \boldsymbol{\sigma} \in \mathbb{R}^{N \times C \times 1 \times 1}$. With these definitions, Adaptive Instance Normalization is computed using the same broadcast-based expression as in Equation (36).

### 4.7 FiLM

The final example is Feature-wise Linear Modulation (FiLM) (Perez et al., 2018), widely used in image generation: $\text{FiLM}(\boldsymbol{F}_{i,c} \mid \gamma_{i,c}, \beta_{i,c}) = \gamma_{i,c} \boldsymbol{F}_{i,c} + \beta_{i,c}$. Here, $\boldsymbol{F}_{i,c}$ denotes the $c$-th channel response of the $i$-th instance, and $\gamma_{i,c}$ and $\beta_{i,c}$ are scaling and shifting weights. The operation is difficult to express in matrix notation, making the expression cumbersome.

With the broadcast product, we can intuitively express FiLM. Let $\mathcal{F} \in \mathbb{R}^{N \times C \times H \times W}$ be a feature volume with the number of instances $N$, $C$ channels, height $H$, width $W$. The weights are represented as $\boldsymbol{\gamma}, \boldsymbol{\beta} \in \mathbb{R}^{N \times C \times 1 \times 1}$. FiLM can be expressed as:

$$\text{FiLM}(\mathcal{F} \mid \boldsymbol{\gamma}, \boldsymbol{\beta}) = \mathcal{F} \boxdot \boldsymbol{\gamma} \boxplus \boldsymbol{\beta}. \tag{37}$$

This formulation removes explicit indexing and places FiLM in the same broadcast-based framework as normalization layers, revealing FiLM as an instance transformation without normalization.

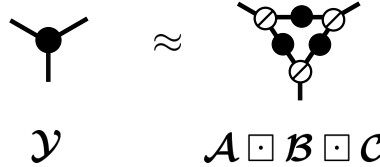

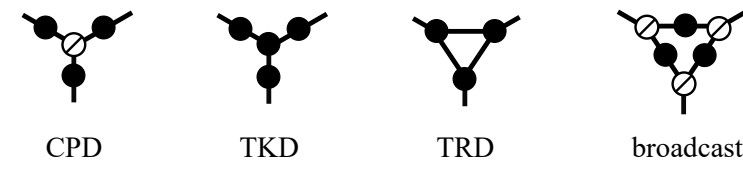

Figure 3: Broadcast decomposition

Figure 4: Graphical notation of canonical polyadic decomposition (CPD), Tucker decomposition (TKD), tensor ring decomposition (TRD), and broadcast decomposition (BD).

## 5 Optimizations

In this section, we delve into the broadcast product in greater detail, particularly in the context of tensor decomposition, and address the new optimization challenges that arise from it, including matrix factorization techniques and their applications in dimensionality reduction.

### 5.1 Least squares

Let us consider three tensors $\boldsymbol{\mathcal{Y}} \in \mathbb{R}^{I \times J \times K}$, $\boldsymbol{\mathcal{A}} \in \mathbb{R}^{I \times J \times 1}$, $\boldsymbol{\mathcal{Z}} \in \mathbb{R}^{1 \times J \times K}$ and the following least squares (LS) problem:

$$\underset{\boldsymbol{\mathcal{A}}}{\text{minimize}} \, ||\boldsymbol{\mathcal{Y}} - \boldsymbol{\mathcal{A}} \boxdot \boldsymbol{\mathcal{Z}}||_F^2, \tag{38}$$

then the solution can be given by

$$\widehat{\boldsymbol{\mathcal{A}}} = \mathcal{P}_3(\boldsymbol{\mathcal{Y}} \boxdot \boldsymbol{\mathcal{Z}}) \boxslash \mathcal{P}_3(\boldsymbol{\mathcal{Z}} \boxdot \boldsymbol{\mathcal{Z}}), \tag{39}$$

where $\mathcal{P}_k(\boldsymbol{\mathcal{X}}) := \boldsymbol{\mathcal{X}} \times_k \mathbf{1}^\top$ performs a sum of the entries of an input tensor along the $k$-th mode. The proof is in Appendix D. This least-squares solution can be easily generalized to $N$-th order tensors by simply changing the axes of $\mathcal{P}$ to match the shape of $\boldsymbol{\mathcal{A}}$ (i.e., summing along the axes with length 1 of $\boldsymbol{\mathcal{A}}$).

### 5.2 Tensor decomposition

We propose a new tensor decomposition called broadcast decomposition (BD), as a proof-of-concept to illustrate the expressive power of the broadcast product operator rather than as a mature method. It is defined via broadcast products as follows:

$$\boldsymbol{\mathcal{Y}} \approx \boldsymbol{\mathcal{A}} \boxdot \boldsymbol{\mathcal{B}} \boxdot \boldsymbol{\mathcal{C}}, \tag{40}$$

where $\boldsymbol{\mathcal{A}}$, $\boldsymbol{\mathcal{B}}$, and $\boldsymbol{\mathcal{C}}$ mutually satisfy the broadcast conditions. For minimizing squared errors $||\boldsymbol{\mathcal{Y}} - \boldsymbol{\mathcal{A}} \boxdot \boldsymbol{\mathcal{B}} \boxdot \boldsymbol{\mathcal{C}}||_F^2$, the alternating least squares (ALS) algorithm can be easily derived using Equation (39). For example, when updating $\boldsymbol{\mathcal{A}}$, set $\boldsymbol{\mathcal{Z}} = \boldsymbol{\mathcal{B}} \boxdot \boldsymbol{\mathcal{C}}$ and make $\mathcal{P}$ correspond to the shape of $\boldsymbol{\mathcal{A}}$. Let us consider the sizes of tensors as $\boldsymbol{\mathcal{A}} \in \mathbb{R}^{I \times J \times 1}$, $\boldsymbol{\mathcal{B}} \in \mathbb{R}^{I \times 1 \times K}$, $\boldsymbol{\mathcal{C}} \in \mathbb{R}^{1 \times J \times K}$, then these update rules[3] can be given by

$$\boldsymbol{\mathcal{A}} \leftarrow \mathcal{P}_3(\boldsymbol{\mathcal{Y}} \boxdot \boldsymbol{\mathcal{B}} \boxdot \boldsymbol{\mathcal{C}}) \oslash \mathcal{P}_3(\boldsymbol{\mathcal{B}} \boxdot \boldsymbol{\mathcal{B}} \boxdot \boldsymbol{\mathcal{C}} \boxdot \boldsymbol{\mathcal{C}}); \tag{41}$$

$$\boldsymbol{\mathcal{B}} \leftarrow \mathcal{P}_2(\boldsymbol{\mathcal{Y}} \boxdot \boldsymbol{\mathcal{A}} \boxdot \boldsymbol{\mathcal{C}}) \oslash \mathcal{P}_2(\boldsymbol{\mathcal{A}} \boxdot \boldsymbol{\mathcal{A}} \boxdot \boldsymbol{\mathcal{C}} \boxdot \boldsymbol{\mathcal{C}}); \tag{42}$$

$$\boldsymbol{\mathcal{C}} \leftarrow \mathcal{P}_1(\boldsymbol{\mathcal{Y}} \boxdot \boldsymbol{\mathcal{A}} \boxdot \boldsymbol{\mathcal{B}}) \oslash \mathcal{P}_1(\boldsymbol{\mathcal{A}} \boxdot \boldsymbol{\mathcal{A}} \boxdot \boldsymbol{\mathcal{B}} \boxdot \boldsymbol{\mathcal{B}}); \tag{43}$$

Here, the axis for the sum operation $\mathcal{P}$ is determined according to the shape of the update tensor. It corresponds to the axis with length 1 of the update tensor. Note that the algorithm will diverge when the denominator includes zero entries. It is safer to avoid zero entries as an initial value. This problem is less likely to occur if we assume $\boldsymbol{\mathcal{Y}}$, $\boldsymbol{\mathcal{A}}$, $\boldsymbol{\mathcal{B}}$, and $\boldsymbol{\mathcal{C}}$ are positive or non-negative tensors. Figures 3 and 4 show tensor networks of BD and other tensor decompositions in graphical notation.

---

[3]More generally $\oslash$ can be replaced with $\boxslash$ if the shapes are different.

**Computational complexity.** The computational cost of BD is $O(IJK)$ per ALS step for a tensor of size $(I, J, K)$, both for the least-squares solution (Equation (39)) and for each factor matrix update. This is comparable to a rank-1 CP decomposition, since one ALS update step for a rank-$R$ CP decomposition costs $O(IJKR) + O(R^3)$, which reduces to $O(IJK)$ at $R = 1$. Other properties such as sensitivity to initialization remain important directions for future work.

Furthermore, the expressive power of the model can be improved by considering the sum of BDs:

$$\boldsymbol{\mathcal{Y}} \approx \sum_{r=1}^{R} \boldsymbol{\mathcal{A}}^{(r)} \boxdot \boldsymbol{\mathcal{B}}^{(r)} \boxdot \boldsymbol{\mathcal{C}}^{(r)}, \tag{44}$$

where $\boldsymbol{\mathcal{A}}^{(r)} \in \mathbb{R}^{I \times J \times 1}$, $\boldsymbol{\mathcal{B}}^{(r)} \in \mathbb{R}^{I \times 1 \times K}$, and $\boldsymbol{\mathcal{C}}^{(r)} \in \mathbb{R}^{1 \times J \times K}$ mutually satisfy the broadcast conditions for each $r \in \{1, 2, \ldots, R\}$. The sum of BDs shown in Equation (44) is an extension of the outer product in CP decomposition (Hitchcock, 1927; Carroll & Chang, 1970; Harshman, 1970) to the broadcast product. For minimizing squared errors $||\boldsymbol{\mathcal{Y}} - \sum_{r=1}^{R} \boldsymbol{\mathcal{A}}^{(r)} \boxdot \boldsymbol{\mathcal{B}}^{(r)} \boxdot \boldsymbol{\mathcal{C}}^{(r)}||_F^2$, the hierarchical ALS (HALS) (Cichocki et al., 2007) can be adapted. The objective function of the sub-problem for the $k$-th component is given by $||\boldsymbol{\mathcal{Y}}_k - \boldsymbol{\mathcal{A}}^{(k)} \boxdot \boldsymbol{\mathcal{B}}^{(k)} \boxdot \boldsymbol{\mathcal{C}}^{(k)}||_F^2$, where $\boldsymbol{\mathcal{Y}}_k := \boldsymbol{\mathcal{Y}} - \sum_{r \neq k} \boldsymbol{\mathcal{A}}^{(r)} \boxdot \boldsymbol{\mathcal{B}}^{(r)} \boxdot \boldsymbol{\mathcal{C}}^{(r)}$, then the update rules can be derived in the same way as Equation (41), Equation (42), and Equation (43).

### 5.3 Difference from conventional TDs

In this section, we provide a preliminary illustration of how BD differs from conventional tensor decompositions (TDs). Note that the tensors we want to approximate may represent some real data or they may represent parameters of a neural network in real applications, but the concrete case study is beyond the scope of this paper. Our goal here is to show, at least in part, that BD has distinct structural properties from other decomposition models. Note that synthetic results below are generated from the BD model and are therefore expected to favor BD by construction; the real-data results should be interpreted as preliminary.

First, we constructed a synthetic tensor $\boldsymbol{\mathcal{W}} \in \mathbb{R}^{32 \times 32 \times 32}$ as follows:

$$\boldsymbol{\mathcal{W}} = \boldsymbol{\mathcal{A}} \boxdot \boldsymbol{\mathcal{B}} \boxdot \boldsymbol{\mathcal{C}} + \sigma \boldsymbol{\mathcal{E}}, \tag{45}$$

where $\boldsymbol{\mathcal{A}} \in \mathbb{R}^{32 \times 32 \times 1}$, $\boldsymbol{\mathcal{B}} \in \mathbb{R}^{32 \times 1 \times 32}$, $\boldsymbol{\mathcal{C}} \in \mathbb{R}^{1 \times 32 \times 32}$ and noise $\boldsymbol{\mathcal{E}} \in \mathbb{R}^{32 \times 32 \times 32}$ are randomly generated, and $\sigma > 0$. We also used a real traffic speed data tensor[4] of size $(32, 21, 24)$ measured every 24 hours at 32 locations over 21 days. The data tensor is treated as the ground truth, and a small amount of Gaussian noise is added to it, which is then treated as the observed data.

Next, we applied CP decomposition (CPD), Tucker decomposition (TKD) (Tucker, 1966; De Lathauwer et al., 2000), tensor-ring decomposition (TRD) (Zhao et al., 2016), and the proposed sum of BDs to a tensor $\boldsymbol{\mathcal{W}}$. The optimization algorithm was applied with various values of the rank parameters, and the signal-to-noise ratio (SNR) of the reconstructed tensors was evaluated by

$$\text{SNR} = 10 \log \frac{||\boldsymbol{\mathcal{W}}_0||_F^2}{||\boldsymbol{\mathcal{W}}_0 - \widehat{\boldsymbol{\mathcal{W}}}||_F^2}, \tag{46}$$

where $\boldsymbol{\mathcal{W}}_0$ is a ground-truth synthetic or real tensor without noise (i.e., $\boldsymbol{\mathcal{W}}_0 = \boldsymbol{\mathcal{A}} \boxdot \boldsymbol{\mathcal{B}} \boxdot \boldsymbol{\mathcal{C}}$ for synthetic) and $\widehat{\boldsymbol{\mathcal{W}}}$ is a reconstructed tensor obtained by TD algorithms. The number of model parameters and SNRs are shown in Figure 5. It can be seen that while the proposed BD succeeds in achieving

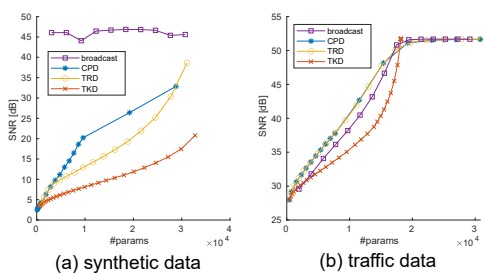

(a) synthetic data      (b) traffic data

Figure 5: Dimensionality reduction of synthetic and traffic tensors: This suggests the existence of a tensor structure that favors broadcast decomposition.

accurate approximation, other low-rank TD models have difficulty achieving efficient approximation in synthetic data. In contrast, the results from real data suggest that the BD is intermediate between CPD/TRD and TKD. Although BD and TDs are similar in the sense that they are compact representations with few parameters, the properties of tensors are significantly different.

---

[4]The data is publicly available at `http://www.openits.cn/openData2/792.jhtml`

# 6 Related Work

## 6.1 Relationship with existing tensor products

The prototype of the broadcast product has been proposed as the penetrating face (PF) product (Slyusar, 1999). The PF product is only defined for a matrix $\boldsymbol{X} \in \mathbb{R}^{I \times J}$ and a tensor $\boldsymbol{\mathcal{Y}} \in \mathbb{R}^{I \times J \times K}$ as

$$\boldsymbol{X} \odot_{\mathrm{PF}} \boldsymbol{\mathcal{Y}} = \boldsymbol{X} \boxdot \boldsymbol{\mathcal{Y}}. \tag{47}$$

This can be regarded as a generalization of the Hadamard product between two matrices and a special case of the broadcast product at the same time. The broadcast product gives an integrated general definition of both the Hadamard product and the PF product.

From the perspective of graphical notation, broadcast product can be thought of as an operation that connects the corresponding modes of two tensors via third-order super-diagonal tensors (see Figure 2). When the third-order super-diagonal tensors are $(1, 1, 1)$ in size, it is equivalent to a simple outer product. The broadcast product is notable for not including an inner-product-like operation (operation involving the $\sum$ symbol) that directly connects the edges of tensors. Its mathematical structure is completely different from that of ordinary tensor products, Einstein products Brazell et al. (2013), and t-products (Zhang et al., 2014; Zhang & Aeron, 2016).

The broadcast product is closely related to the Khatri-Rao (KR) product. The KR product is defined between two matrices with the same number of columns $\boldsymbol{A} = [\boldsymbol{a}_1, \ldots, \boldsymbol{a}_R] \in \mathbb{R}^{I \times R}$ and $\boldsymbol{B} = [\boldsymbol{b}_1, \ldots, \boldsymbol{b}_R] \in \mathbb{R}^{J \times R}$:

$$\boldsymbol{B} \odot_{\mathrm{KR}} \boldsymbol{A} = [\boldsymbol{b}_1 \otimes \boldsymbol{a}_1, \ldots, \boldsymbol{b}_R \otimes \boldsymbol{a}_R] \in \mathbb{R}^{IJ \times R}, \tag{48}$$

where $\otimes$ is the Kronecker product. The following relationship holds:

$$\boldsymbol{B} \odot_{\mathrm{KR}} \boldsymbol{A} = \mathrm{fold}_3(\tilde{\boldsymbol{\mathcal{A}}} \boxdot \tilde{\boldsymbol{\mathcal{B}}})^{\top}, \tag{49}$$

where $\tilde{\boldsymbol{\mathcal{A}}} \in \mathbb{R}^{I \times 1 \times R}$ and $\tilde{\boldsymbol{\mathcal{B}}} \in \mathbb{R}^{1 \times J \times R}$ are reshaped tensors of $\boldsymbol{A}$ and $\boldsymbol{B}$. In other words, the KR product and the broadcast product have the same mathematical structure because $\mathrm{fold}_3(\cdot)$ and $\cdot^{\top}$ are just tensor reshaping operations. Noting that the KR product is defined only for matrices, the broadcast product can therefore be regarded as a generalization of the KR product in a broader sense.

## 6.2 Notation for describing complex mathematical expressions

Some papers that aim for accurate descriptions have already introduced the concept of the broadcast product, such as Wang et al. (2022) and You et al. (2024). Unlike us, they have not discussed the mathematical properties. Also, Wang et al. (2022) used $\otimes$ as the symbol for the broadcast product, but $\otimes$ is generally used for the Kronecker product. This confusion can be avoided by using our $\boxdot$. See Appendix E for a comprehensive list of symbol conflicts for element-wise multiplication.

The einops notation (Rogozhnikov, 2022), the einx notation (Fervers et al., 2026), and the detailed Transformer description (Phuong & Hutter, 2022) serve as valuable references for clear mathematical descriptions. On the practical side, `TensorLy` (Kossaifi et al., 2019) is a modern library for tensor processing.

## 6.3 Named tensor notation

Named Tensor Notation (Chiang et al., 2023) shares the most similar motivation with ours. By explicitly naming each axis, Named Tensor Notation allows complex tensor operations to be expressed concisely and code-like. For example, the masking operation discussed in Section 1 is described as follows:

$$Y = X \odot B, \quad \text{where } Y, X \in \mathbb{R}^{\texttt{height} \times \texttt{width} \times \texttt{chans}}, \ B \in \mathbb{R}^{\texttt{height} \times \texttt{width}} \tag{50}$$

Broadcasting is automatically performed when the axes do not match (Def. 6 in Chiang et al. (2023)), so broadcast products can be represented with the above simple expression. Named Tensor Notation has several valuable properties, such as omitting axis ordering and hiding axis lengths.

For comparison, the same masking operation in our notation is:

$$\boldsymbol{\mathcal{Y}} = \boldsymbol{\mathcal{X}} \boxdot \boldsymbol{B}, \quad \boldsymbol{\mathcal{X}} \in \mathbb{R}^{H \times W \times 3}, \; \boldsymbol{B} \in \mathbb{R}^{H \times W}. \tag{51}$$

Shapes are specified numerically, and the $\boxdot$ operator handles the broadcasting. Because the resulting expression is standard tensor notation, it is directly substitutable into mathematical inequalities and identities (as illustrated in Sections 1 and 3).

**Trade-offs.** The two approaches have complementary strengths. Our broadcast product is fully compatible with standard mathematical notation: results can be substituted directly into equations, no axis-naming scheme is required, and it connects naturally to existing tensor algebra such as Frobenius-norm properties, least-squares solutions, and decompositions (see Sections 3 and 5). The main cost is that shapes must be specified explicitly for each operand. Named Tensor Notation, on the other hand, offers richer expressiveness (axis naming, reductions, and advanced indexing in a concise, code-like style) and leaves axis ordering implicit, reducing notational overhead for complex multi-axis operations. Its limitations are that a consistent axis-naming scheme must be maintained throughout the paper, and that results cannot be directly substituted into standard mathematical expressions.

**When to prefer each.** When a broadcast operation must be connected to standard tensor algebra (for instance, when deriving norm inequalities or least-squares solutions), our notation provides a cleaner formulation. It is also preferable when the same broadcast pattern recurs across multiple equations or when broadcasting is a structural feature of the model, as in broadcast decomposition. When the goal is a concise description of complex multi-axis operations, or when axis ordering should be left implicit, Named Tensor Notation may be preferable. When a single broadcast alignment must be specified in isolation, explicit shape annotation (e.g., $\boldsymbol{B}^{\square} \in \mathbb{R}^{H \times W \times 3}$) can also be more readable than either. Named Tensor Notation and our broadcast product represent different approaches with the same motivation; developing a unifying theory would be a direction for future work.

## 7 Conclusion

We redefined the broadcast operation commonly used in scientific computing libraries. The broadcast product enhances the accuracy of mathematical expressions in machine learning and other fields. Additionally, we contextualized this product within tensor decomposition, highlighting its potential for developing new types of tensor decompositions. Demonstration code for the examples in this paper is available at `https://github.com/matsui528/broadcast_product_demo`.

### Acknowledgments

This work was partially supported by the Japan Society for the Promotion of Science (JSPS) KAKENHI under Grant 23K28109.

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

Figure 6: The broadcast product of a third-order tensor and a matrix Equation (28): $\boldsymbol{\mathcal{X}} \boxdot \boldsymbol{Y} = \boldsymbol{\mathcal{X}} \odot \text{fold}_1(\mathbf{1}_K^\top \otimes \boldsymbol{Y}) = \text{fold}_3(\boldsymbol{X}_{(3)}\text{diag}(\text{vec}(\boldsymbol{Y})))$

Yuxin Wu and Kaiming He. Group normalization. In *Proceedings of the European Conference on Computer Vision (ECCV)*, 2018.

Tatsuya Yokota. Very basics of tensors with graphical notations: Unfolding, calculations, and decompositions. *arXiv preprint arXiv:2411.16094*, 2024.

Kaichao You, Guo Qin, Anchang Bao, Meng Cao, Ping Huang, Jiulong Shan, and Mingsheng Long. Efficient convbn blocks for transfer learning and beyond. In *Proceedings of the International Conference on Learning Representations (ICLR)*, 2024.

Zemin Zhang and Shuchin Aeron. Exact tensor completion using t-svd. *IEEE Transactions on Signal Processing*, 65(6):1511–1526, 2016.

Zemin Zhang, Gregory Ely, Shuchin Aeron, Ning Hao, and Misha Kilmer. Novel methods for multilinear data completion and de-noising based on tensor-SVD. In *Proceedings of the IEEE Conference on Computer Vision and Pattern Recognition (CVPR)*, pp. 3842–3849, 2014.

Qibin Zhao, Guoxu Zhou, Shengli Xie, Liqing Zhang, and Andrzej Cichocki. Tensor ring decomposition. *arXiv preprint arXiv:1606.05535*, 2016.

## A Proof of Theorem 3.1

We prove Theorem 3.1. For two tensors $\boldsymbol{\mathcal{X}}, \boldsymbol{\mathcal{Y}}$ that satisfy the broadcast condition,

$$
\begin{aligned}
\|\boldsymbol{\mathcal{X}} \boxdot \boldsymbol{\mathcal{Y}}\|_F &= \|\boldsymbol{\mathcal{X}}_\square \odot \boldsymbol{\mathcal{Y}}_\square\|_F & \text{(Equation (20))} & \quad (52) \\
&\leq \|\boldsymbol{\mathcal{X}}_\square\|_F \cdot \|\boldsymbol{\mathcal{Y}}_\square\|_F & \text{(Cauchy–Schwarz for the Hadamard product)} & \quad (53) \\
&= \|\boldsymbol{\mathcal{X}}\|_F \cdot \|\boldsymbol{\mathcal{Y}}\|_F. & \text{(Definition of marginalization)} & \quad (54)
\end{aligned}
$$

Thus, the claim follows.

Here, Cauchy–Schwarz for the Hadamard product (Horn & Johnson, 1994) means that for any tensors $\boldsymbol{\mathcal{A}}, \boldsymbol{\mathcal{B}}$ of the same shape, $\|\boldsymbol{\mathcal{A}} \odot \boldsymbol{\mathcal{B}}\|_F \leq \|\boldsymbol{\mathcal{A}}\|_F \cdot \|\boldsymbol{\mathcal{B}}\|_F$. This is obtained by considering $\boldsymbol{A} = \text{diag}(\boldsymbol{\mathcal{A}})$ and $\boldsymbol{B} = \text{diag}(\boldsymbol{\mathcal{B}})$, and substituting them into $\|\boldsymbol{AB}\|_F \leq \|\boldsymbol{A}\|_F \cdot \|\boldsymbol{B}\|_F$ (the sub-multiplicativity of the Frobenius norm).

## B Visualizing Third-order Tensors and Matrices

Figure 6 visualizes Equation (28), the broadcast product of a third-order tensor and a matrix. Here, the Kronecker product is denoted by $\boldsymbol{X} \otimes \boldsymbol{Y}$. The mode-$n$ unfolding $\boldsymbol{X}_{(n)} \in \mathbb{R}^{I_n \times \prod_{k \neq n} I_k}$ rearranges all entries of $\boldsymbol{\mathcal{X}}$ into a matrix whose rows are indexed by the $n$-th axis (Kolda & Bader, 2009; Yokota, 2024); for example, $\boldsymbol{X}_{(1)} \in \mathbb{R}^{I \times JK}$. Its inverse is denoted $\text{fold}_n(\cdot)$, so that $\text{fold}_n(\boldsymbol{X}_{(n)}) = \boldsymbol{\mathcal{X}}$.

(a) unfold and fold

(b) proof of main property in graphical notation

Figure 7: Proof of property Equation (28) using graphical notation. (a) Unfolding-and-folding is an identity mapping, and it can be applied to proof in (b). In addition, the outer product of two super-diagonal tensors and their unfolding results in another super-diagonal tensor. Then, we can obtain $\boldsymbol{\mathcal{X}} \boxdot \boldsymbol{Y} = \mathrm{fold}_3(\boldsymbol{X}_{(3)}\mathrm{diag}(\mathrm{vec}(\boldsymbol{Y})))$ in graphical notation.

Here, $\mathbf{1}_K^\top \otimes \boldsymbol{Y} \in \mathbb{R}^{I \times JK}$ represents $\boldsymbol{Y}$ repeated $K$ times horizontally. By $\mathrm{fold}_1$, the shape is aligned with that of $\boldsymbol{\mathcal{X}}$. Also, one can express this computation by multiplying the matrix $\boldsymbol{X}_{(3)} \in \mathbb{R}^{K \times IJ}$ by the diagonal matrix $\mathrm{diag}(\mathrm{vec}(\boldsymbol{Y})) \in \mathbb{R}^{IJ \times IJ}$. Here, we first unfold each frontal slice of $\boldsymbol{\mathcal{X}}$ and arrange them to form a matrix. Then, for each row, we take the product with the unfolded $\boldsymbol{Y}$. Finally, the result is folded back into its original shape using $\mathrm{fold}_3$.

In addition, the broadcast product of a third-order tensor and a matrix shown in Equation (28) can be analyzed using graphical notation in Figure 7. Here, unfolding and folding operators are denoted by half circles with three lines. The analysis is based on the topological equivalence of graphs and properties of operators of unfolding, folding with super-diagonal tensors. For example, operations of unfolding-and-folding and folding-and-unfolding are identity mappings (see Figure 7a). The outer product of two super-diagonal tensors and their unfolding results in another super-diagonal tensor (see Figure 7b).

## C  Properties of broadcast sum, difference, and division

The broadcast difference can be expressed using the broadcast sum through the following trivial transformation. For tensors $\boldsymbol{\mathcal{X}}$ and $\boldsymbol{\mathcal{Y}}$ satisfying the broadcast condition,

$$\boldsymbol{\mathcal{X}} \boxminus \boldsymbol{\mathcal{Y}} = \boldsymbol{\mathcal{X}} \boxplus (-\boldsymbol{\mathcal{Y}}). \tag{55}$$

Similarly, the broadcast division can be expressed using the broadcast product through the following transformation. For tensors $\boldsymbol{\mathcal{X}}$ and $\boldsymbol{\mathcal{Y}}$ satisfying the broadcast condition (with $\boldsymbol{\mathcal{Y}}$ containing no zero elements),

$$\boldsymbol{\mathcal{X}} \boxslash \boldsymbol{\mathcal{Y}} = \boldsymbol{\mathcal{X}} \boxdot (\mathbf{1} \oslash \boldsymbol{\mathcal{Y}}), \tag{56}$$

where $\mathbf{1}$ is a tensor of the same shape as $\boldsymbol{Y}$, with all elements equal to 1. Therefore, focusing on the properties of the broadcast sum and product is sufficient.

Here, we introduce the basic properties of broadcast sum. First, Equations (12), (13) and (16) also hold if $\boxdot$ and $\odot$ are replaced with $\boxplus$ and $+$, respectively.

**Scalar**: For any tensor $\boldsymbol{\mathcal{X}}$ and a scalar $y$, we obtain

$$\boldsymbol{\mathcal{X}} \boxplus y = \boldsymbol{\mathcal{X}} + y\mathbf{1}, \tag{57}$$

where $\mathbf{1}$ is an all-one tensor having the same shape as $\boldsymbol{\mathcal{X}}$. This equation may seem extremely trivial initially, but it is important. Adding a constant to all tensor elements is the most basic form of broadcasting, and in `numpy` it is written as a simple addition like `X + y`. Unfortunately, many papers directly write this in mathematical notation as $X + y$. Such a description is incorrect and must be written as $\boldsymbol{\mathcal{X}} + y\mathbf{1}$ as shown above. In addition, there are frequent mistakes where this operation is misunderstood, and the identity matrix $\boldsymbol{I}$ is incorrectly used, resulting in expressions like $\boldsymbol{X} + y\boldsymbol{I}$.

**Vector and vector**: For vectors $\boldsymbol{x} \in \mathbb{R}^I$ and $\boldsymbol{y} \in \mathbb{R}^J$ of different lengths, we obtain

$$\boldsymbol{x} \boxplus \boldsymbol{y}^\top = \boldsymbol{x}\boldsymbol{1}_J^\top + \boldsymbol{1}_I \boldsymbol{y}^\top \in \mathbb{R}^{I \times J}. \tag{58}$$

**Matrix and vector**: For a matrix $\boldsymbol{X} \in \mathbb{R}^{I \times J}$, vectors $\boldsymbol{y} \in \mathbb{R}^I$ and $\boldsymbol{z} \in \mathbb{R}^J$, the following holds:

$$\boldsymbol{X} \boxplus \boldsymbol{y} = \boldsymbol{X} + [\boldsymbol{y}|\boldsymbol{y}|\dots|\boldsymbol{y}] = \boldsymbol{X} + \boldsymbol{y}\boldsymbol{1}_J^\top. \tag{59}$$

$$\boldsymbol{X} \boxplus \boldsymbol{z}^\top = \boldsymbol{X} + \begin{bmatrix} \boldsymbol{z}^\top \\ \vdots \\ \boldsymbol{z}^\top \end{bmatrix} = \boldsymbol{X} + \boldsymbol{1}_I \boldsymbol{z}^\top. \tag{60}$$

## D  Proof of Least Squares Solution

### D.1  Case of the third-order tensors

Here, we show the derivation of the LS solution Equation (39):

$$\widehat{\boldsymbol{\mathcal{A}}} = \underset{\boldsymbol{\mathcal{A}}}{\operatorname{argmin}} ||\boldsymbol{\mathcal{Y}} - \boldsymbol{\mathcal{A}} \boxdot \boldsymbol{\mathcal{Z}}||_F^2 \tag{61}$$

for $\boldsymbol{\mathcal{Y}} \in \mathbb{R}^{I \times J \times K}$, $\boldsymbol{\mathcal{A}} \in \mathbb{R}^{I \times J \times 1}$, and $\boldsymbol{\mathcal{Z}} \in \mathbb{R}^{1 \times J \times K}$. Let us put $\boldsymbol{Y}_j := \boldsymbol{\mathcal{Y}}_{:j:} \in \mathbb{R}^{I \times K}$, $\boldsymbol{a}_j := \boldsymbol{\mathcal{A}}_{:j1} \in \mathbb{R}^I$ and $\boldsymbol{z}_j := \boldsymbol{\mathcal{Z}}_{1j:} \in \mathbb{R}^K$, the squared errors can be transformed as

$$||\boldsymbol{\mathcal{Y}} - \boldsymbol{\mathcal{A}} \boxdot \boldsymbol{\mathcal{Z}}||_F^2 = \sum_{j=1}^J ||\boldsymbol{Y}_j - \boldsymbol{a}_j \boldsymbol{z}_j^\top||_F^2. \tag{62}$$

Then the solution of $\boldsymbol{a}_j$ can be independently obtained by

$$\widehat{\boldsymbol{a}}_j = \underset{\boldsymbol{a}_j}{\operatorname{argmin}} ||\boldsymbol{Y}_j - \boldsymbol{a}_j \boldsymbol{z}_j^\top||_F^2 = \boldsymbol{Y}_j \boldsymbol{z}_j (\boldsymbol{z}_j^\top \boldsymbol{z}_j)^{-1} \tag{63}$$

for each $j \in \{1, 2, \dots, J\}$. Since $(i, j, 1)$-th entry of $\widehat{\boldsymbol{\mathcal{A}}}$ corresponds to $i$-th entry of $\widehat{\boldsymbol{a}}_j$, we have

$$\widehat{a}_{ij1} = \widehat{\boldsymbol{a}}_j(i) = \left(\sum_{k=1}^K y_{ijk} z_{jk}\right) \left(\sum_{k=1}^K z_{jk}^2\right)^{-1} \iff \widehat{\boldsymbol{\mathcal{A}}} = \mathcal{P}_3(\boldsymbol{\mathcal{Y}} \boxdot \boldsymbol{\mathcal{Z}}) \boxslash \mathcal{P}_3(\boldsymbol{\mathcal{Z}} \boxdot \boldsymbol{\mathcal{Z}}). \tag{64}$$

### D.2  Case of the $N$-th order tensors

Let us consider $N$-th order tensors $\boldsymbol{\mathcal{W}} \in \mathbb{R}^{D_1 \times D_2 \times \cdots \times D_N}$, $\boldsymbol{\mathcal{H}} \in \mathbb{R}^{F_1 \times F_2 \times \cdots \times F_N}$ and $\boldsymbol{\mathcal{X}} \in \mathbb{R}^{\max(D_1, F_1) \times \max(D_2, F_2) \times \cdots \times \max(D_N, F_N)}$, then the problem can be written by

$$\widehat{\boldsymbol{\mathcal{W}}} = \underset{\boldsymbol{\mathcal{W}}}{\operatorname{argmin}} ||\boldsymbol{\mathcal{X}} - \boldsymbol{\mathcal{W}} \boxdot \boldsymbol{\mathcal{H}}||_F^2. \tag{65}$$

Without loss of generality, we can reduce the problem with $N$-th order tensors $(\boldsymbol{\mathcal{X}}, \boldsymbol{\mathcal{W}}, \boldsymbol{\mathcal{H}})$ in Equation (65) to the problem with third-order tensors $(\boldsymbol{\mathcal{Y}}, \boldsymbol{\mathcal{A}}, \boldsymbol{\mathcal{Z}})$ in Equation (61). Since $\boldsymbol{\mathcal{W}}$ and $\boldsymbol{\mathcal{H}}$ satisfy the broadcast condition, the $N$ modes can be divided into three categories:

$$\mathcal{L} = \{ n \mid D_n > 1, F_n = 1 \}, \tag{66}$$
$$\mathcal{S} = \{ n \mid D_n = F_n \}, \tag{67}$$
$$\mathcal{R} = \{ n \mid D_n = 1, F_n > 1 \}. \tag{68}$$

$\mathcal{L}$ is the set of broadcasting modes for $\boldsymbol{\mathcal{H}}$, corresponding to the first mode of $\boldsymbol{\mathcal{Z}}$ in Equation (61). $\mathcal{S}$ is the set of non-broadcasting modes, corresponding to the second mode in Equation (61). $\mathcal{R}$ is the set of broadcasting

modes for $\boldsymbol{\mathcal{W}}$, corresponding to the third mode of $\boldsymbol{\mathcal{A}}$ in Equation (61). Then, we convert $N$-th order tensors to third-order tensors based on $(\mathcal{L}, \mathcal{S}, \mathcal{R})$ using mode permutation and tensor unfolding as follows:

$$\boldsymbol{\mathcal{Y}} = \text{unfold}_{(I,J,K)}\text{permute}_{(\mathcal{L},\mathcal{S},\mathcal{R})}(\boldsymbol{\mathcal{X}}) \in \mathbb{R}^{I \times J \times K}, \tag{69}$$

$$\boldsymbol{\mathcal{A}} = \text{unfold}_{(I,J,1)}\text{permute}_{(\mathcal{L},\mathcal{S},\mathcal{R})}(\boldsymbol{\mathcal{W}}) \in \mathbb{R}^{I \times J \times 1}, \tag{70}$$

$$\boldsymbol{\mathcal{Z}} = \text{unfold}_{(1,J,K)}\text{permute}_{(\mathcal{L},\mathcal{S},\mathcal{R})}(\boldsymbol{\mathcal{H}}) \in \mathbb{R}^{1 \times J \times K}, \tag{71}$$

where $I = \prod_{n \in \mathcal{L}} D_n$, $J = \prod_{n \in \mathcal{S}} D_n$, and $K = \prod_{n \in \mathcal{R}} F_n$. The LS solution of third-order tensors $\widehat{\boldsymbol{\mathcal{A}}}$ can be obtained by Equation (64). By converting $\widehat{\boldsymbol{\mathcal{A}}}$ back to an $N$-th order tensor, the solution can be obtained as follows:

$$\widehat{\boldsymbol{\mathcal{W}}} = \text{permute}_{(\mathcal{L},\mathcal{S},\mathcal{R})}^{-1}\text{unfold}_{(I,J,1)}^{-1}\left(\widehat{\boldsymbol{\mathcal{A}}}\right) = \mathcal{P}_{\mathcal{R}}(\boldsymbol{\mathcal{X}} \boxdot \boldsymbol{\mathcal{H}}) \boxslash \mathcal{P}_{\mathcal{R}}(\boldsymbol{\mathcal{H}} \boxdot \boldsymbol{\mathcal{H}}), \tag{72}$$

where $\text{permute}_{(\mathcal{L},\mathcal{S},\mathcal{R})}^{-1}$ and $\text{unfold}_{(I,J,1)}^{-1}$ are inverse of $\text{permute}_{(\mathcal{L},\mathcal{S},\mathcal{R})}$ and $\text{unfold}_{(I,J,1)}$, and $\mathcal{P}_{\mathcal{R}}(\cdot)$ is a sum operation along the modes in $\mathcal{R}$. For example, let be

$$\boldsymbol{\mathcal{X}} \in \mathbb{R}^{10 \times 20 \times 30 \times 40 \times 50 \times 60},$$
$$\boldsymbol{\mathcal{W}} \in \mathbb{R}^{10 \times 20 \times 1 \times 40 \times 50 \times 1},$$
$$\boldsymbol{\mathcal{H}} \in \mathbb{R}^{10 \times 1 \times 30 \times 1 \times 50 \times 60},$$
$$\mathcal{L} = \{2, 4\}, \mathcal{S} = \{1, 5\}, \mathcal{R} = \{3, 6\},$$

then the permutation operation outputs

$$\tilde{\boldsymbol{\mathcal{X}}} = \text{permute}_{(\{2,4\},\{1,5\},\{3,6\})}(\boldsymbol{\mathcal{X}}) \in \mathbb{R}^{20 \times 40 \times 10 \times 50 \times 30 \times 60},$$
$$\tilde{\boldsymbol{\mathcal{W}}} = \text{permute}_{(\{2,4\},\{1,5\},\{3,6\})}(\boldsymbol{\mathcal{W}}) \in \mathbb{R}^{20 \times 40 \times 10 \times 50 \times 1 \times 1},$$
$$\tilde{\boldsymbol{\mathcal{H}}} = \text{permute}_{(\{2,4\},\{1,5\},\{3,6\})}(\boldsymbol{\mathcal{H}}) \in \mathbb{R}^{1 \times 1 \times 10 \times 50 \times 30 \times 60},$$

the unfolding operation outputs

$$\boldsymbol{\mathcal{Y}} = \text{unfold}_{(800,500,1800)}(\tilde{\boldsymbol{\mathcal{X}}}) \in \mathbb{R}^{800 \times 500 \times 1800},$$
$$\boldsymbol{\mathcal{A}} = \text{unfold}_{(800,500,1)}(\tilde{\boldsymbol{\mathcal{W}}}) \in \mathbb{R}^{800 \times 500 \times 1},$$
$$\boldsymbol{\mathcal{Z}} = \text{unfold}_{(1,500,1800)}(\tilde{\boldsymbol{\mathcal{H}}}) \in \mathbb{R}^{1 \times 500 \times 1800},$$

and the sum operation outputs

$$\mathcal{P}_{\{3,6\}}(\boldsymbol{\mathcal{X}} \boxdot \boldsymbol{\mathcal{H}}) = (\boldsymbol{\mathcal{X}} \boxdot \boldsymbol{\mathcal{H}}) \times_3 \mathbf{1}^\top \times_6 \mathbf{1}^\top \in \mathbb{R}^{10 \times 20 \times 1 \times 40 \times 50 \times 1},$$
$$\mathcal{P}_{\{3,6\}}(\boldsymbol{\mathcal{H}} \boxdot \boldsymbol{\mathcal{H}}) = (\boldsymbol{\mathcal{H}} \boxdot \boldsymbol{\mathcal{H}}) \times_3 \mathbf{1}^\top \times_6 \mathbf{1}^\top \in \mathbb{R}^{10 \times 1 \times 1 \times 1 \times 50 \times 1}.$$

# E   Conflicts of Mathematical Symbols

Here we discuss the issue of symbols for element-wise multiplication. As shown in Table 1, the symbols used to represent element-wise multiplication (Hadamard product) are very diverse and most of them conflict with other mathematical operations.

Since the symbol $\boxdot$ for the broadcast product proposed in this paper does not conflict and can also be used for the Hadamard product, it may solve such problems. In addition, we can represent both the Hadamard product and the broadcast product using only one symbol $\boxdot$. A promising set of conflict-free notations would be given as follows.

Table 1: List of symbols used for element-wise multiplication

| Symbol | Usage / conflict |
|---|---|
| ∘ | Used in (Slyusar, 1999; Bernstein, 2009; Theodoridis, 2020). Conflicts with the outer product in (Cichocki et al., 2009; Kolda & Bader, 2009; Strang, 2019). |
| ⊙ | Used in (Markovsky, 2012; Goodfellow et al., 2016; Kochenderfer, 2019). Conflicts with the Khatri–Rao product in (Cichocki et al., 2009; Kolda & Bader, 2009; Strang, 2019). |
| ⊛ | Used in (Cichocki et al., 2009). Conflicts with convolution in (Jain, 1989; Strang, 2019). |
| ∗ | Used in (Kolda & Bader, 2009). Conflicts with convolution (Strang, 2019) and the t-product (Kernfeld et al., 2015). |
| .∗ | Used in (Golub & Van Loan, 2013; Cichocki et al., 2009; Strang, 2019). No conflicts reported. |
| ⊡ (ours) | No conflicts reported (including Hadamard product). |

Table 2: Input commands for the broadcast operators.

| Operation | Symbol | LaTeX package | LaTeX input | PowerPoint equation input |
|---|---|---|---|---|
| Broadcast product | ⊡ | `amssymb` | `\boxdot` | `\boxdot` + Space |
| Broadcast sum | ⊞ | `amssymb` | `\boxplus` | `\boxplus` + Space |
| Broadcast difference | ⊟ | `amssymb` | `\boxminus` | `\boxminus` + Space |
| Broadcast division | ⊘ | `stmaryrd` | `\boxslash` | Copy and paste `U+29C4` directly |

- outer product ∘
- Kronecker product ⊗
- Khatri-Rao product ⊙
- Hadamard product (with broadcast option) ⊡

- element-wise division (with broadcast option) ⊘
- convolution ⊛
- t-product ∗

Note that this paper does not follow the above notation set, but it was necessary because of the definition of the broadcast product using the Hadamard product and discussing their relationship.

## F  Typesetting the Broadcast Operators

The broadcast operators introduced in this paper are represented by standard Unicode mathematical symbols. Their input commands in LaTeX and Microsoft PowerPoint are summarized in Table 2.

In LaTeX, ⊡, ⊞, and ⊟ are available through the `amssymb` package, whereas ⊘ is available through the `stmaryrd` package. In Microsoft PowerPoint, ⊡, ⊞, and ⊟ can be entered by inserting an equation box and typing the corresponding command followed by a space; PowerPoint interprets these through its UnicodeMath / Math AutoCorrect input mechanism. Unfortunately, ⊘ is not supported by this mechanism and must be inserted by directly copying and pasting `U+29C4`.

## G  Translation from numpy/Julia

The proposed broadcast product is almost identical to `numpy`'s broadcast operation, allowing `A * B` in `numpy` to be represented as $\mathcal{A} \boxdot \mathcal{B}$ in equations. However, there is a difference in how cases are handled when the tensor orders differ, but the broadcasting condition is satisfied.

The key difference lies in the shape-normalization convention (Definition 1): our definition follows the **F-convention** (trailing singletons are appended), whereas `numpy` follows the **C-convention** (leading singletons are prepended)[5]. For instance, $\mathbb{R}^{2\times3}$ is identified with $\mathbb{R}^{2\times3\times1}$ under the F-convention, but with $\mathbb{R}^{1\times2\times3}$ under the C-convention. Below, we show some examples of Python codes.

---

[5]https://numpy.org/doc/stable/user/basics.broadcasting.html

As in the following example, if we explicitly specify the tensors' shapes including modes of length one, all computations work as expected. We can directly translate `A * B` in numpy to $\mathcal{A} \boxdot \mathcal{B}$ in equations.

```
A = np.random.rand(2, 3, 4)    # A.shape == (2, 3, 4), i.e., A ∈ ℝ^{2×3×4}
B = np.random.rand(2, 3, 1)    # B.shape == (2, 3, 1), i.e., B ∈ ℝ^{2×3×1}

# (ℝ^{2×3×4}, ℝ^{2×3×1}) satisfies the broadcast condition. Thus we can compute A ⊡ B
A * B    # OK
```

However, if we don't explicitly write down the axis of length one, numpy's behavior is in the exact opposite way to our definition as follows.

```
C = np.random.rand(2, 3)    # C.shape == (2, 3), i.e., C ∈ ℝ^{2×3}
D = np.random.rand(3, 4)    # D.shape == (3, 4), i.e., D ∈ ℝ^{3×4}

# F-convention identifies ℝ^{2×3} with ℝ^{2×3×1}, so A ⊡ C can be computed.
# However, numpy's C-convention identifies ℝ^{2×3} with ℝ^{1×2×3}, so A * C fails.
A * C    # ValueError: operands could not be broadcast together with shapes (2,3,4) (2,3)

# F-convention identifies ℝ^{3×4} with ℝ^{3×4×1}, so A ⊡ D cannot be computed.
# However, numpy's C-convention identifies ℝ^{3×4} with ℝ^{1×3×4}, so A * D succeeds.
A * D    # OK
```

If confusion happens when writing the broadcast product, we recommend explicitly defining the shape even for the axis with length one. For example, an image with a single channel can be written as $\mathbb{R}^{H \times W \times 1}$.

In fact, since `Julia` follows the F-convention (Definition 1), its behavior is exactly the same as our broadcast product. In `Julia`, the broadcast product is expressed by `.*` as follows.

```
A = rand(2, 3, 4)    # size(A) == (2, 3, 4), i.e., A ∈ ℝ^{2×3×4}
B = rand(2, 3, 1)    # size(B) == (2, 3, 1), i.e., B ∈ ℝ^{2×3×1}
# (ℝ^{2×3×4}, ℝ^{2×3×1}) satisfies the broadcast condition. Thus we can compute A ⊡ B
A .* B    # OK

C = rand(2, 3)    # size(C) == (2, 3), i.e., C ∈ ℝ^{2×3}
D = rand(3, 4)    # size(D) == (3, 4), i.e., D ∈ ℝ^{3×4}
# Under the F-convention, ℝ^{2×3} is identified with ℝ^{2×3×1}, so A ⊡ C can be computed.
A .* C    # OK
# Under the F-convention, ℝ^{3×4} is identified with ℝ^{3×4×1}, so A ⊡ D cannot be computed.
A .* D    # ERROR: DimensionMismatch("arrays could not be broadcast to a common size; ..
```

