# OpenReview forum: "Broadcast Product: Redefining Shape-aligned Element-wise Multiplication and Beyond"
_TMLR — Accepted by TMLR_

### Review · Reviewer_pSAp · 2026-03-04

**Summary Of Contributions:**

Authors describe conflicts between code and math notation caused by usage of broadcasting operator.
They introduce shape-broadcasting ("boxed") operations into math notation, and describe its properties.

Additionally, a "marginalization" (akin to torch.norm(x, dim=..., keepdim=True)) is introduced.

Authors rewrite multiple common normalizations using this notation, and suggest a tensor decomposition.

**Additional Comments:**

for reference: Penrose notation lost battle to Einstein notation in physics (and not used anywhere else AFAIK).
And I'm puzzled by cameo of Penrose notation in this paper.

Proposed notation also doesn't help with derivations: formulas still 61-63 use usual matrix notation.

**Audience:**

Yes

**Audience Explanation:**

Topic is important: tensor notation is lacking, we indeed need better tools to describe broadcasting in algorithms.

I don't like suggestion in this paper, and find supportive evidence very weak.

**Broader Impact Concerns:**

No ethical concerns

**Claims And Evidence:**

No

**Claims Explanation:**

I have strong complains about proposed notation as it introduces discrepancies with operations in frameworks, and the arguments made in favor of this notation (see details below).

Experimental validation (section 5.3) can't be considered as a supportive argument.

**Requested Changes:**

1. broadcasting in numpy/torch/etc aligns right-to-left, not left-to-right. It is inacceptable to take common term and give it different contradictory meaning. Call it F-broadcasting if you want (F because fortran). This doesn't help with math<>code duality, and adds more confusion.
I see this mentioned in Appendix F, it does not make any excuse.

Also: axis numbering (used in formulas) starts from one. While technically correct, it shows weak place of notation.

Minor: Fold/unfold is never explained, and is not a term in the community (anymore).

2. sup-box for aligned tensors (8) would suffice, I don't see need to introduce any additional operations like broadcasted sum/div/prod. We don't use them in code because we want smaller intermediate tensors.
Sup-box has its own problems (e.g. one can't take a part of equation or merge equations, because sup-box depends on all marked tensors), but it is simple enough and is just one new operation.
In particular, in (5.2) I can immediately see where (41)-(43) come from if sup-box was used instead of operations.
Even more practical way would be to introduce operation like:
(x', y', z') = broadcast(x, y, z)


3.  authors write several normalizations using same formula (36), finding it better than explicit (35).
I can't agree with this: if by looking at formula I can't tell which normalization was used, formula did not convey necessary information. In normalization cases, I'd strongly prefer "canonical" notation to proposed.

4. experimental support: proposed benchmark strongly favors selected decomposition, as generates examples within same domain. Can't be considered an argument.

5. broadcast decomposition: multiple decompositions can be framed as "broadcast", however used shapes for components are crucial, and paper does not perform this analysis.

---

> ### Author Response · Authors · 2026-04-02
>
> We thank the reviewer for the careful and constructive feedback. We agree that the paper can be substantially improved in its terminology, self-contained exposition, and clarity of the scope of the empirical claims. Below, we address each point and indicate the corresponding revisions we will make.
>
> ## (1) right-to-left and left-to-right
>
> We thank the reviewer for pointing out this essential issue. Our intention is not to introduce confusion, but rather to propose one approach to resolving the confusion that already exists. Specifically,
> in NumPy, a right-to-left convention is used (i.e., $\mathbb{R}^{1 \times 3}$ is identified with $\mathbb{R}^3$), whereas in mathematical notation, a left-to-right convention is used ($\mathbb{R}^{3 \times 1}$ is identified with $\mathbb{R}^3$). We intended to make this distinction explicit. Mathematical notation and framework conventions are often implicitly mixed in the literature, leading to ambiguous tensor shapes. Our notation aims to clarify this difference and provide a unified representation.
>
> We found the reviewer's suggestion of adding a prefix such as "F-", as in F-broadcasting, to be very clear and helpful. We will revise the manuscript and explicitly state at the beginning of page 4, in the description of tensor identification, that mathematical notation and Julia follow the "F-format." In contrast, NumPy and PyTorch follow the "C-format." We will also add a concrete example and state explicitly which format is used in each subsequent definition. Thank you very much again for the reviewer's valuable suggestion, which helps improve the paper.
>
> ## (2) Fold/unfold
>
> We agree that the manuscript is currently not self-contained on this point. We will add concise definitions of fold/unfold [Sec 4.4 in Yokota2024] in the main text.
>
> [Yokota, 2024] Yokota, "Very Basics of Tensors with Graphical Notations: Unfolding, Calculations, and Decompositions", arXiv (2024)
>
>
>
> ## (3) Is sup-box sufficient?
>
> We agree that in some cases the superscript-box notation is sufficient and may even be more intuitive, especially when the goal is simply to indicate explicit duplication of a tensor. Our motivation for $\boxdot$  is different: it attaches broadcasting semantics to an operation, which lets a single expression cover multiple shape configurations without rewriting the formula case by case.
>
> For example, when considering the process of applying weights $A$ element-wise to an image $X$, writing $X \boxdot A$ allows $X$ to be either color (3 channels) or grayscale (1 channel), and $A$ can also be either 3-channel or 1-channel.
>
> We will add to the manuscript that a style using only the sup-box is also possible, and include a comparison with the broadcasting product formulation.

---

> ### Author Response · Authors · 2026-04-02
>
> ## (4) Normalization
>
> The reviewer's concern is entirely reasonable. We do not intend to enforce the use of the proposed notation. As the reviewer pointed out, explicitly listing all elements ensures complete rigor, and we fully agree that such expressions can be clearer in several cases.
>
> The intuitive advantage of our unified notation (Eq. 36) is twofold. First, its results can be directly reused in other expressions. For example, Eq. 36 can be substituted into another operation as is. Second, generalization can reveal underlying relationships. For instance, FiLM (Eq. 37) can be viewed as Eq. 36 without normalization.
>
> ## (5) Weakness of experiments
>
> We agree with the reviewer that the current empirical section is limited. The proposed broadcasting decomposition should be viewed as a proof-of-concept on synthetic data, not as a mature decomposition method or as evidence of broad empirical superiority. We will revise the text accordingly and make clear that the experiments serve only to illustrate that the proposed broadcasting product can induce a new family of decomposition formulations.
>
> ## (6) Penrose notation
>
> We agree that Penrose notation may be unfamiliar to some readers outside tensor-network communities. Our intent in using it is to make shape structure and contractions visually explicit. We will better motivate this choice, cite representative uses in computational physics [Ran, 2020] and machine learning [Stoudenmire2016, Cichocki2016, Cichocki2017, Li2023, Fernandez2025., and add a brief explanatory paragraph so that readers can follow the notation without prior familiarity.
>
> - [Ran, 2020] Ran, Shi-Ju, Emanuele Tirrito, Cheng Peng, Xi Chen, Luca Tagliacozzo, Gang Su, and Maciej Lewenstein. Tensor network contractions: methods and applications to quantum many-body systems. Springer Nature, 2020.
> - [Stoudenmire, 2016] Stoudenmire, Edwin, and David J. Schwab. "Supervised learning with tensor networks." Advances in Neural Information Processing Systems 29 (2016).
> - [Cichocki, 2016] Cichocki, Andrzej, et al. "Tensor networks for dimensionality reduction and large-scale optimization part 1 low-rank tensor decompositions." Foundations and Trends in Machine Learning 9.4-5 (2016): 249-429.
> - [Cichocki, 2017] Cichocki, Andrzej, et al. "Tensor networks for dimensionality reduction and large-scale optimizations part 2 applications and future perspectives." Foundations and Trends in Machine Learning 9.6 (2017): 431-673.
> - [Li, 2023] Li, Chao, et al. "Alternating local enumeration (tnale): Solving tensor network structure search with fewer evaluations." International Conference on Machine Learning. PMLR, 2023.
> - [Fernandez, 2025] Núñez Fernández, Yuriel, et al. "Learning tensor networks with tensor cross interpolation: New algorithms and libraries." SciPost Physics 18.3 (2025): 104.

---

> ### Author Response · Authors · 2026-04-22
>
> We thank the reviewer for the detailed and direct feedback. We have addressed all of the concerns:
>
> **Right-to-left vs. left-to-right broadcasting (F-convention vs. C-convention).** This was the most structurally impactful suggestion in the review: the F-convention/C-convention distinction is now formalized as Definition 1 at the start of Sec. 2, which clarifies the mathematical foundation of the entire paper. We introduced a formal Definition of F-convention (Fortran-style, trailing singletons appended, used in mathematical notation and Julia) and C-convention (C-style, leading singletons prepended, used in numpy and PyTorch). All subsequent definitions reference this explicitly. We now state clearly that our paper adopts the F-convention throughout, and Appendix F (numpy/Julia translation) has been rewritten to use this terminology. *(Sec. 2, Def. 1, at the very start of Sec. 2; Appendix F)*
>
> **fold/unfold definition.** We added a concise definition of mode-$n$ unfolding and its inverse $\mathrm{fold}_n$ directly in the main text, citing Kolda & Bader (2009) and Yokota (2024). *(Sec. 3.3 (Properties of lower-order tensors), "Third-order tensor and matrix" paragraph; also Appendix B)*
>
> **When is sup-box sufficient?** We added a paragraph explicitly acknowledging that the sup-box notation ($X^\square \odot Y^\square$) is always mathematically correct and may be preferable when the broadcast structure of a specific expression is the focus. We explain that $\boxdot$ is preferable when the same broadcast pattern recurs across multiple formulas, as it attaches broadcasting semantics to the operation rather than the operands. *(Sec. 2)*
>
> **Weakness of the empirical section.** We added a real traffic speed tensor experiment and explicitly note in the text that the synthetic benchmark favors BD by construction. The framing has been changed from "demonstrating BD's superiority" to "providing a preliminary illustration of how BD differs structurally from conventional TDs." *(Sec. 5.3, first two paragraphs and Figure 5)*
>
> **Computational complexity of BD.** We added a paragraph stating that the cost is $O(IJK)$ per ALS step, comparable to rank-1 CP decomposition, and note that other properties such as sensitivity to initialization remain future work. *(Sec. 5.2, "Computational complexity" paragraph, after the ALS update rules)*
>
> **Penrose notation motivation.** We added a paragraph motivating the use of Penrose graphical notation with references to its standard use in tensor-network communities in computational physics and machine learning. *(Sec. 3.4 (Graphical notations), at the end of the subsection)*

---

### Review · Reviewer_g58G · 2026-03-19

**Summary Of Contributions:**

This paper introduces a new notation for the _broadcast product_ of two tensors, closely resembling the `*` operator in NumPy and PyTorch. Some properties of the notation are proved, and a new kind of tensor decomposition, _broadcast decomposition_, is introduced that is especially convenient to state in the new notation.

**Additional Comments:**

Section 1 before 1.1

mode: perhaps make a comment that in array libraries this is called a “dimension” or an “axis”. I agree with your choice of the word “mode” but it may be unfamiliar to readers.
the domain of the numbers -> the underlying field
Khatri-Rao product: citation and/or definition needed

Section 2

The alignment (“add ones to the end of a shape”) rule should be part of Definition 1, not in a “Note that”. If you consider an n-dimensional vector and an n \times 1 matrix to be the same, then the properties on page 6 don’t make sense.

Section 3

mutually satisfy -> pairwise satisfy
correctly defined -> well-defined
as discussed in Equation (1) -> as discussed in connection with Equation (1)
In Definition 3 and elsewhere, what does “focusing on elements” mean? Perhaps it’s better to say “using index notation” or “using element-wise notation”?
Why is “marginalization” called that? Is it related at all to marginalization of probability distributions?
The MATLAB-style colon (:) notation is not defined, and I find it strange to let a variable equal :.
super-diagonal: define
I’m not sure what the notation in 3.4 adds to the paper. Moreover, doesn’t this notation makes a distinction between covariant and contravariant modes, which you don’t use elsewhere?

**Audience:**

Yes

**Audience Explanation:**

Nearly everyone in TMLR's audience has used broadcasting in code, and probably a good number have experienced the awkwardness of not having broadcasting in standard mathematical notation.

**Claims And Evidence:**

Yes

**Claims Explanation:**

The broadcast product notation is reasonable, the properties appear to be correct, and broadcast decomposition is reasonable. My only reservation vis-a-vis the criterion of "accurate, convincing, and clear evidence" is that the experiment in section 5.3 seems designed to favor broadcast decomposition. However, the limitations of this experiment are very clearly stated.

**Requested Changes:**

Many of the examples in Section 4 are also use-cases for named tensor notation (Chiang et al, 2023), which is discussed in 6.3, but the advantage of the broadcast notation over named tensor notation is not clearly argued. I think a more careful comparison of these notations would be helpful.

Broadcast decomposition (Section 5) is interesting, and is conveniently written using broadcast products, but of course it doesn’t depend on this notation in any essential way. I wonder if it's possible to make a stronger case that these two ideas belong in the same paper.

---

> ### Author Response · Authors · 2026-04-02
>
> We thank the reviewer for their valuable comments. Below, we respond to each question in turn.
>
> ## (1) Named Tensor Notation
>
> We thank the reviewer for the comment. As the reviewer pointed out, Named Tensor Notation addresses a concern similar to ours. In the revised manuscript, we will include a more detailed comparative discussion as follows.
>
> The main advantage of the proposed broadcast product is its strong compatibility with standard mathematics. The results of our broadcasting operations are fully consistent with conventional mathematical expressions, which allows them to be directly substituted into other equations. In contrast, Named Tensor Notation introduces a distinct representational framework that differs from standard mathematics. Usually, the results cannot be easily substituted into standard mathematical expressions.
>
> On the other hand, Named Tensor Notation offers rich functionalities, including explicit axis naming, differentiation support, reductions, advanced indexing, and more. Therefore, there is a clear trade-off between the two approaches.
>
> ## (2) Can broadcast decomposition be written without broadcast notation?
>
> We agree that broadcast decomposition can be written without introducing a new symbol. However, our point is not merely that the notation makes the expressions shorter. Rather, the decomposition is defined by taking the broadcast product as a primitive structured interaction, in the same way that CP decomposition is built from outer products and matrix factorizations are built from matrix products.
>
> In the revision, we will clarify this connection more explicitly: Section 5 is included not as an unrelated application, but as evidence that the proposed operation is mathematically productive. In particular, the least-squares formulation and the graphical interpretation in Section 5 rely on the structural view of the broadcast product developed in Sections 2-3. We will revise the introduction to make this dependency clearer.
>
> ## (3) Mode
>
> We thank the reviewer for this suggestion. We will replace all descriptions of "mode" with "axis." In cases where "mode" is standard terminology in the tensor decomposition literature (e.g., nth-mode tensor-matrix product), we will include appropriate clarifications.
>
> ## (4) "add ones to the end of the shape"
>
> We agree that this point should be included in the formal definition rather than just as a note. Reviewer pSAp also recommended revising this aspect. In the revision, we will first establish a clear shape-normalization rule that adds trailing singleton modes as necessary, which we will refer to as "Fortran-format" (F-format), as suggested by Reviewer pSAp. Additionally, we will define "C-format" for the leading singleton modes. We will clearly outline the broadcast condition by explicitly referencing the F-format. We believe this modification addresses the reviewer's concern and clarifies the entire mathematical formulation.
>
> ## (5) Minor revisions and definitions
>
> We agree that "focusing on elements" is imprecise. We will replace this phrasing with "index notation" (or "element-wise notation," where appropriate) throughout the paper. We will also add a formal definition for ":".
>
> We thank the reviewer again for the comment on marginalization. The term was chosen because of its resemblance to marginalization in probability mass functions; however, we acknowledge that its appropriateness may be subject to interpretation. In probability, marginalization removes one axis while preserving the sum (i.e., the $\ell_1$ norm) of the tensor. In contrast, our definition also removes one axis while preserving the Frobenius norm. Although the preserved quantities differ, we chose the term based on this structural similarity. That said, to avoid potential confusion, we will revise the terminology to "Frobenius-norm marginalization."
>
> ## (6) Super-diagonal
>
> We introduced a graphical notation using super-diagonal tensors to capture the underlying mathematical structure effectively. As illustrated in Fig. 7, such graphical notations help derive properties of the broadcast product intuitively. Furthermore, as shown in Section 5.2, this representation enables a graphical formulation of broadcast decomposition, which may help connect our work to the field of tensor networks. In addition, in Section 6.1, we discussed the relationship between the broadcast product and the Khatri-Rao product. This relationship can also be understood more intuitively through graphical notations (see [Yokota, 2024] for graphical notations of the Khatri-Rao product).
>
> [Yokota, 2024] Yokota, "Very Basics of Tensors with Graphical Notations: Unfolding, Calculations, and Decompositions", arXiv (2024)

---

> ### Author Response · Authors · 2026-04-22
>
> We thank the reviewer for their supportive review and the many helpful detailed suggestions. We addressed all of the requested changes:
>
> **Named Tensor Notation comparison.** We substantially expanded Sec. 6.3 with a side-by-side example showing the same masking operation in both notations, followed by a "Trade-offs" paragraph and a "When to prefer each" paragraph. We explain that our notation is fully compatible with standard mathematical expressions (substitutable into inequalities, norm identities, and least-squares derivations), while Named Tensor Notation offers richer expressiveness and implicit axis ordering. *(Sec. 6.3, second half)*
>
> **Motivation for including broadcast decomposition.** We added a sentence at the start of Sec. 5.2 explicitly stating that broadcast decomposition is a proof-of-concept intended to illustrate the mathematical productivity of the broadcast product operator, and is not a mature decomposition method. *(Sec. 5.2, first sentence)*
>
> **Alignment rule in Definition 1.** We introduced a formal Definition of F-convention and C-convention before the broadcast condition, and incorporated the singleton-appending rule into the broadcast condition definition itself. The old "Note that" sentence has been removed. *(Sec. 2, Def. 1 (F-convention and C-convention) and Def. 2 (Broadcast Condition), at the start of Sec. 2)*
>
> **"Focusing on elements" -> index notation.** We replaced all occurrences of "focusing on elements" with "using index notation" (or "element-wise notation" where appropriate). *(Sec. 2 (broadcast product definition) and Sec. 3.2)*
>
> **Super-diagonal: formal definition added.** We added a one-sentence formal definition of super-diagonal tensor. *(Sec. 3.4 (Graphical notations), just before the example)*
>
> **"Marginalization" renamed to "Frobenius-norm Marginalization".** We renamed the subsection and definition, and added an explanatory sentence connecting the name to the analogy with probability marginalization (both reduce one axis while preserving a norm). *(Sec. 3.2, subsection heading and Def. 4)*
>
> **Colon notation defined.** We added a sentence citing the MATLAB/Julia convention (Golub & Van Loan, 2012) when the colon notation is first introduced. *(Sec. 3.2, just before Def. 4)*
>
> **"Mutually satisfy" -> "pairwise satisfy".** Fixed in the Associativity property. *(Sec. 3.1, Associativity paragraph)*
>
> **"Domain" -> "underlying field".** Fixed in the Notation paragraph. *(Sec. 1, Notation paragraph, last sentence)*
>
> **Khatri-Rao product: citation and definition added.** We added the definition (column-wise Kronecker products of matrices with the same number of columns) and a citation to Kolda & Bader (2009) at first use. *(Sec. 1.1 (Examples), Example 2)*

---

### Review · Reviewer_A9t8 · 2026-03-31

**Summary Of Contributions:**

The paper proposes a formal broadcast product operator to make implicit broadcasting mathematically explicit in tensor expressions, especially for common ML formulas such as masking, normalization, and FiLM. It gives a clean definition, derives several algebraic properties, connects the operator to existing tensor constructions and graphical notation, and sketches optimization procedures for least-squares fitting and a new "broadcast decomposition." The presentation is generally clear and the motivation around mismatches between code semantics and mathematical notation is legitimate. However, the strongest claims go beyond what the evidence currently supports: the optimization and decomposition sections are preliminary, the empirical validation is limited to synthetic data generated from the proposed model class, and the novelty relative to prior tensor products and notation systems is not fully pinned down.

**Additional Comments:**

The paper is easiest to accept as a rigorous notation and formalization paper, not yet as a convincing tensor-methods paper. If the authors reposition it more tightly around formal semantics and notation, the contribution becomes clearer. If they want to keep the decomposition angle central, they need substantially stronger evidence and a more careful comparison to existing tensor models.

**Audience:**

Yes

**Audience Explanation:**

There is a real audience for this paper within ML theory, tensor methods, and researchers who care about mathematical clarity in ML writing. The paper addresses a genuine friction point: many papers and implementations rely on implicit broadcasting while writing equations that are formally invalid. Making those semantics explicit could improve rigor, exposition, and possibly some forms of tensor modeling. That said, the interest is somewhat specialized. The notation contribution is more likely to appeal to readers focused on tensor algebra, decomposition, or formal specification than to the broader TMLR readership. The work would become substantially more compelling if it demonstrated that the formalism leads to new analyses, better optimization procedures, or materially better modeling outcomes beyond syntactic cleanup.

**Claims And Evidence:**

No

**Claims Explanation:**

The definitional and algebraic parts are mostly well supported: the paper clearly states the broadcast condition, defines the operator constructively, and provides useful examples showing why informal use of the Hadamard product can be mathematically invalid. Those parts make the notation proposal credible. The issue is that several broader claims are supported only weakly. The "real-world examples" section does not validate a new method; it mainly rewrites known formulas in the proposed notation. The optimization/decomposition story is much less convincing: the least-squares result is stated with proof deferred to the appendix, and the experimental section uses synthetic tensors generated exactly from the proposed broadcast structure, making it unsurprising that the proposed decomposition outperforms conventional tensor decompositions on that benchmark. There is no evaluation on real tensors, no ablation on optimization stability or identifiability, little discussion of failure cases, and no evidence that the notation or decomposition improves an actual ML workflow. So the paper supports the claim that broadcast-aware notation can be formalized cleanly, but it does not yet convincingly support stronger claims about practical decomposition value or broader impact on machine learning methodology.

**Requested Changes:**

1. (Critical) Strengthen the empirical section with evaluations beyond synthetic data generated from the proposed model. At minimum, test broadcast decomposition on real tensors or realistic parameter tensors and compare against well-tuned baselines under matched parameter budgets.
2. (Critical) Sharpen the novelty claim relative to prior work. The paper acknowledges the PF product, Khatri-Rao connections, and named tensor notation, but the boundary between "new notation," "generalized existing construction," and "new decomposition model" remains too blurry.
3. (Critical) Narrow or qualify claims about practical usefulness. As written, the paper suggests broad implications for optimization and dimensionality reduction, but the evidence currently supports only a preliminary proof of concept.
4. (Minor) Add more analysis of the proposed decomposition itself: identifiability, optimization landscape, computational complexity, sensitivity to initialization, and cases where it should or should not outperform standard tensor decompositions.
5. (Minor) Improve discussion of when the proposed symbol should be preferred over alternatives such as named tensor notation or explicit broadcasting annotations, since the notation burden for readers is part of the tradeoff.

---

> ### Author Response · Authors · 2026-04-06
>
> We sincerely thank the reviewer for the constructive comments. As the reviewer correctly noted, the current manuscript did not sufficiently distinguish between the contribution in notation/formalization and the preliminary application to broadcast decomposition. In the revised manuscript, we will clarify that the primary contributions are limited to: (i) the proposal of an explicit notation for broadcast-aware elementwise operations, and (ii) the rigorous definition and organization of their basic properties. By contrast, broadcast decomposition will be repositioned as an exploratory proof-of-concept. Below, we respond to each comment in turn.
>
> ## (1) Evaluation of the decomposition
>
> The reviewer is correct that the current experiments rely only on synthetic data and therefore provide limited support for the claims. We conducted an additional experiment on a real tensor of traffic speeds [1], and compared the proposed method with CP, Tucker, and Tensor-ring decompositions under matched parameter budgets (#params $\sim 8 \times 10^3$). The simplified results of the experiment are as follows.
>
> | Method                         | SNR [dB] |
> | ------------------------------ | -------- |
> | Tucker decomposition           | 34.2     |  (8322)
> | CP decomposition               | 37.8     | (7700)
> | Tensor-ring decomposition      | 37.9     | (7700)
> | Broadcast decomposition (Ours) | 36.1     | (7776)
>
> These experimental results show that our BD decomposition achieves higher accuracy than Tucker, although it is inferior to CP and Tensor-ring. We will add the complete plots of these experiments to the final manuscript. Investigating to find effective applications of this new BD decomposition model would require extensive exploratory research and is outside the scope of this study. Therefore, we will simply report that "the BD possesses different properties" would be our contribution to the part of decomposition.
>
> If there is any data that the reviewers would like us to include in the final version, please let us know. We must conduct the necessary experiments and include them in the camera-ready version.
>
> [1] http://www.openits.cn/openData2/792.jhtml
>
>
> ## (2) Novelty claim
> The reviewer rightly pointed out that the novelty claim was not sufficiently precise. In the revised manuscript, we will define the contributions more clearly as follows:
>
> - Notation/formalization contribution: We propose a notation for explicitly representing broadcast-aware elementwise operations.
> - Analytical contribution: Under this notation, we provide a systematic definition of the broadcast product, summarize its basic properties, and clarify its relationship to existing operations.
>
> Regarding the tensor decomposition component, we will explicitly treat it as a proof-of-concept and clarify that it should be regarded as a preliminary result rather than a central contribution. In addition, the Related Work section will be expanded to more explicitly discuss the relationship to the PF product and the Khatri-Rao product, as well as the differences from existing broadcast-aware descriptions. This revision will make the distinction between introducing a new symbol and generalizing existing constructions clearer. We appreciate the reviewer’s comments, which will significantly improve the clarity and positioning of the paper.
>
> ## (3) Practical usefulness
>
> The reviewer also correctly noted that the claims regarding practical usefulness were too strong. In the revised manuscript, we will narrow these claims to the level that is currently supported. Specifically, we will limit the practical significance of the proposed method to its role as a notation that explicitly and accurately expresses broadcasting operations in mathematical form. By contrast, broader claims regarding effectiveness for optimization or dimensionality reduction are currently supported only by preliminary evidence, and the manuscript will be revised accordingly.

---

> > ### Author Response · Authors · 2026-04-06
> >
> > ## (4) Analysis of the decomposition
> > Although tensor decomposition is not positioned as a primary contribution, we agree with the reviewer that additional analysis of this aspect would strengthen the paper. Accordingly, we will add a discussion of the tensor decomposition viewpoint in the revised manuscript.
> >
> > In particular, as shown in Eq. (39), the computational complexity of obtaining the least-squares solution for a tensor of size $(I, J, K)$ is $O(IJK)$. Similarly, the computational cost of each ALS step for updating a factor matrix is also $O(IJK)$. By comparison, when optimizing a rank-$R$ CP decomposition by ALS, the cost of one update step is roughly $O(IJKR) + O(R^3)$. Therefore, the computational cost of broadcast decomposition may be regarded as being on the same order as that of rank-1 CP decomposition.
> >
> > For other properties, such as sensitivity to initialization, we will explicitly state that these remain important directions for future work.
> >
> > ## (5) When is the proposed notation preferable?
> > The reviewer made an important point that, once a new symbol is introduced, the trade-off with existing notations should be discussed explicitly. In the revised manuscript, we will compare the proposed notation with (a) named-tensor notation and (b) explicit broadcasting annotations, and we will add a discussion clarifying when the proposed notation offers a useful balance between conciseness and precision, and when existing notations may remain easier to read.
> >
> > More specifically, we will clarify the intended distinction as follows: when only a single shape alignment must be specified, explicit annotation may be more readable; however, when the same type of broadcast pattern appears repeatedly, the proposed notation can provide a more concise and systematic description.

---

> ### Author Response · Authors · 2026-04-22
>
> We sincerely thank the reviewer for their detailed and constructive review. We have addressed all of the concerns in the revised manuscript. Specifically, we made the following changes:
>
> **[Critical] Empirical evaluation beyond synthetic data.** We added an experiment on a real traffic speed tensor (size 32x21x24, publicly available data) and compared BD against CP, Tucker, and Tensor-Ring decompositions under matched parameter budgets. The results show that BD achieves higher accuracy than Tucker decomposition, while CP and Tensor-Ring perform better, confirming that BD has distinct structural properties rather than broad superiority. We also explicitly note in the text that the synthetic benchmark is expected to favor BD by construction. *(Sec. 5.3, first paragraph and Figure 5)*
>
> **[Critical] Sharpened novelty claim.** We restructured the contributions in the introduction into three clearly separated bullets: (1) Notation/formalization contribution, (2) Analytical contribution (algebraic properties, relationships to existing operations), and (3) Proof-of-concept decomposition. We explicitly state that BD is a preliminary result and not a central contribution. *(Sec. 1, contributions list)*
>
> **[Critical] Narrowed practical claims.** We removed the overly optimistic sentences about decomposition performance from the paragraph before the contributions list ("Finally, we demonstrate that this decomposition yields lower reconstruction error..." and "...they indicate the potential of broadcast products for dimensionality reduction and other applications."). We also revised the abstract to reposition the paper as primarily a notation and formalization contribution. *(Abstract; Sec. 1, paragraph before contributions list)*
>
> **[Minor] Analysis of the decomposition.** We added a paragraph on computational complexity: the cost of BD is $O(IJK)$ per ALS step for a tensor of size (I, J, K), comparable to rank-1 CP decomposition. We note that other properties such as sensitivity to initialization remain directions for future work. *(Sec. 5.2, "Computational complexity" paragraph)*
>
> **[Minor] When to prefer the proposed notation.** We expanded the Related Work section with a detailed comparison to Named Tensor Notation, including "Trade-offs" and "When to prefer each" paragraphs. This discussion clarifies when our notation offers advantages (e.g., when connecting to standard tensor algebra or when the same broadcast pattern recurs) versus when Named Tensor Notation or explicit sup-box annotation is preferable. *(Sec. 6.3, second half)*

---

### Author Response · Authors · 2026-04-22

Dear Reviewers,

We have uploaded a revised version of the manuscript addressing the comments raised during the review process. Please let us know if any further clarification or revisions would be helpful.

Best regards,

The Authors

---

### Decision · Action_Editor_z6pb · 2026-05-16

**Recommendation:** Accept as is

**Audience:**

Yes

**Audience Explanation:**

Broadcasting is of interest to almost all TMLR readers who touch code at some point, and this paper is arguing for a notation that they think will be less error-prone and more closely connect math with computation.

**Claims And Evidence:**

Yes

**Claims Explanation:**

This is a paper about notation, so it's a bit unusual, but the authors describe the notation thoroughly, and the reviewers find this acceptable.